# Hybridization alters the shape of the genotypic fitness landscape, increasing access to novel fitness peaks during adaptive radiation

Austin H Patton[1,2], Emilie J Richards[1,2], Katelyn J Gould[3], Logan K Buie[3], Christopher H Martin[1,2]*

[1]Department of Integrative Biology, University of California, Berkeley, Berkeley, United States; [2]Museum of Vertebrate Zoology, University of California, Berkeley, Berkeley, United States; [3]Department of Biology, University of North Carolina, Chapel Hill, United States

**Abstract** Estimating the complex relationship between fitness and genotype or phenotype (i.e. the adaptive landscape) is one of the central goals of evolutionary biology. However, adaptive walks connecting genotypes to organismal fitness, speciation, and novel ecological niches are still poorly understood and processes for surmounting fitness valleys remain controversial. One outstanding system for addressing these connections is a recent adaptive radiation of ecologically and morphologically novel pupfishes (a generalist, molluscivore, and scale-eater) endemic to San Salvador Island, Bahamas. We leveraged whole-genome sequencing of 139 hybrids from two independent field fitness experiments to identify the genomic basis of fitness, estimate genotypic fitness networks, and measure the accessibility of adaptive walks on the fitness landscape. We identified 132 single nucleotide polymorphisms (SNPs) that were significantly associated with fitness in field enclosures. Six out of the 13 regions most strongly associated with fitness contained differentially expressed genes and fixed SNPs between trophic specialists; one gene (*mettl21e*) was also misexpressed in lab-reared hybrids, suggesting a potential intrinsic genetic incompatibility. We then constructed genotypic fitness networks from adaptive alleles and show that scale-eating specialists are the most isolated of the three species on these networks. Intriguingly, introgressed and de novo variants reduced fitness landscape ruggedness as compared to standing variation, increasing the accessibility of genotypic fitness paths from generalist to specialists. Our results suggest that adaptive introgression and de novo mutations alter the shape of the fitness landscape, providing key connections in adaptive walks circumventing fitness valleys and triggering the evolution of novelty during adaptive radiation.

*For correspondence:
chmartin@berkeley.edu

Competing interest: The authors declare that no competing interests exist.

## Editor's evaluation

This study reports on the inference of the evolutionary trajectory of two specialist species that evolved from one generalist species. The process of speciation is explained as an adaptive process and the changing genetic architecture of the process is analyzed in great detail. The genomic dataset is big and the inference from it is solid. The authors reach the conclusion that introgression and de novo mutations played a major role in this adaptive process.

**eLife digest** One of the main drivers of evolution is natural selection, which is when organisms better adapted to their environment are more likely to survive and reproduce. A common metaphor to explain this process is a landscape covered in peaks and valleys: the peaks represent genetic combinations or traits with high evolutionary fitness, while the valleys represent those with low fitness.

As a population evolves and its environment changes, it moves among these peaks taking small steps across the landscape. However, there is a limit to how far an organism can travel in one leap. So, what happens when they need to cross a valley of low fitness to get to the next peak? To address this question, Patton et al. studied three young species of pupfish that recently evolved from a common ancestor and co-habit the same environment in the Caribbean.

Patton et al. sequenced whole genomes of each new species and used this to build a genotypic fitness landscape, a network linking neighboring genotypes which each have a unique fitness value that was measured during field experiments. This revealed that most of the paths connecting the different species passed through valleys of low fitness. But there were rare, narrow ridges connecting each species.

Next, Patton et al. found that new mutations as well as genetic variations that arose from mating with pupfish on other Caribbean islands altered genetic interactions and changed the shape of the fitness landscape. Ultimately, this significantly increased the accessibility of fitness peaks by both adding more ridges and decreasing the lengths of paths, expanding the realm of possible evolutionary outcomes.

Understanding how fitness landscapes change during evolution could help to explain where new species come from. Other researchers could apply the same approach to estimate the genotypic fitness landscapes of other species, from bacteria to vertebrates. These networks could be used to visualize the complex fitness landscape that connects all lifeforms on Earth.

## Introduction

First conceptualized by Sewell Wright in 1932, the adaptive landscape describes the complex relationship between genotype or phenotype and fitness (*Wright, 1932*). The landscape is a concept, a metaphor, and an empirical measurement that exerts substantial influence over all evolutionary dynamics (*Gavrilets, 1997*; *Pigliucci and Müller, 2010*; *Svensson and Calsbeek, 2012*; *Fragata et al., 2019*; *Fear and Price, 1998*). Fitness landscapes were originally depicted as high-dimensional networks spanning genotypic space in which each genotype is associated with fitness (*Wright, 1932*). *Simpson, 1944*, later described phenotypic evolution of populations through time on a rugged landscape, in which isolated clusters of fitness peaks represent 'adaptive zones' relative to adjacent regions of low fitness (*Kauffman and Levin, 1987*). Lande and Arnold formalized the analysis of selection and estimation of phenotypic fitness landscapes (*Lande and Arnold, 1983*; *Arnold et al., 2001*; *Arnold, 2003*), leading to empirical studies of fitness landscapes in numerous systems (*Schluter and Grant, 1984*; *Schluter, 1988*; *Hendry et al., 2009*; *Beausoleil et al., 2019*; *Benkman, 2003*; *Martin and Wainwright, 2013a*; *Martin and Gould, 2020*). Fitness surfaces are also central components of speciation models and theory (*Gavrilets, 2004*; *Turelli et al., 2001*; *Servedio and Boughman, 2017*).

A central focus of fitness landscape theory is the characterization of the shape of the fitness landscape. Theoretical and empirical studies frequently attempt to describe its topography, such as quantifying the number of fitness peaks, one component of landscape ruggedness that affects the predictability of evolution (*Fragata et al., 2019*; *Bank et al., 2016*; *Wright, 1931*; *Aita et al., 2001*). Importantly, the existence of multiple peaks and valleys on the fitness landscape implies epistasis for fitness, or non-additive effects on fitness resulting from genotypic interactions (*Wright, 1931*; *Whitlock et al., 1995*; *Poelwijk et al., 2007*; *Poelwijk et al., 2011*). Fitness epistasis reduces the predictability of evolution because the resultant increase in the number of peaks increases the number of viable evolutionary outcomes (*Kauffman and Levin, 1987*; *Neidhart et al., 2014*). Increasing fitness epistasis also increases landscape ruggedness, thus reducing the probability of converging on any one fitness peak and ultimately diversifying potential evolutionary outcomes (*Kauffman and Levin, 1987*; *Neidhart et al., 2014*).

This leads to a fundamental concept in fitness landscape theory: Not all genotypic pathways are evolutionarily accessible (*Fragata et al., 2019*; *Poelwijk et al., 2007*; *Weinreich et al., 2006*; *Ferretti et al., 2018*; *Franke et al., 2011*; *de Visser and Krug, 2014*; *Smith, 1970*; *Ogbunugafor, 2020*). In large populations, paths through genotype space that monotonically increase in fitness at each mutational step are favored over alternatives with neutral or deleterious steps (*Fisher, 1930*). These accessible genotypic paths can be considered adaptive walks under Fisher's geometric model, by which adaptation proceeds toward a phenotypic optimum via additive mutations of small phenotypic effect (*Fisher, 1930*; *Orr, 2005*). On rugged landscapes as originally envisioned by *Wright, 1931*, greater numbers of peaks (i.e. the ruggedness) increase the mean length of potential adaptive walks to any one fitness optimum, while decreasing the length of accessible paths to the nearest peak. Ultimately, this leads to a decrease in the probability that any one fitness optimum is reached. Simultaneously, increasing landscape ruggedness decreases the length of adaptive walks to the nearest local optimum, owing to the corresponding increase in peak density.

There are a growing number of experimental studies of adaptive walks in nature, including the evolution of toxin resistance in monarch butterflies (*Karageorgi et al., 2019*), alcohol tolerance in *Drosophila* (*Siddiq et al., 2017*; *Siddiq and Thornton, 2019*), and host-shift in aphids (*Singh et al., 2020*). Likewise, the accessibility of genotypic fitness networks has now been explored in numerous microbial systems, including the evolution of antibiotic resistance (*Weinreich et al., 2006*), metabolism (*Peng et al., 2018*), citrate exploitation (*Blount et al., 2008*), and glucose limitation in *Escherichia coli* (*Khan et al., 2011*), and adaptation to salinity in yeast via evolution of heat shock protein *Hsp90* (*Bank et al., 2016*). However, these studies are still limited to the investigation of specific coding substitutions and their effects on fitness in laboratory environments. *Nosil et al., 2020* estimated genotypic fitness networks for *Timema* stick insects based on a field experiment. Similarly, this study focused on a single large-effect locus underlying dimorphic coloration between ecotypes. These studies represent significant advances, but extension of fitness landscape theory to empirical systems including multiple species remains an underexplored area of future research at the intersection of micro- and macroevolution. Such studies can provide insight into the topography of fitness landscapes in natural systems, the accessibility of interspecific adaptive walks, and ultimately the predictability of evolution.

One promising system for estimating fitness landscapes is a recent adaptive radiation of *Cyprinodon* pupfishes endemic to San Salvador Island, Bahamas (*Martin and Wainwright, 2013a*; *Martin and Gould, 2020*; *Martin and Wainwright, 2013b*; *Martin, 2016*). This radiation is comprised of two trophic specialists, a molluscivore (durophage: *Cyprinodon brontotheroides*) and a scale-eater (lepidophage: *Cyprinodon desquamator*), derived from a Caribbean-wide generalist (*Cyprinodon variegatus*) which also coexists in the same habitats. These three species all occur in sympatry in the hypersaline lakes of San Salvador Island, Bahamas (*Figure 1a*). Found in the benthic littoral zone of each lake, all three species forage within the same benthic microhabitat; indeed, no habitat segregation has been observed in 14 years of field studies. Originating less than 10,000 years ago (based on geological age estimates for the lakes: *Turner et al., 2008*), the functional and trophic novelty harbored within this radiation is the product of exceptional rates of craniofacial morphological evolution (*Martin and Wainwright, 2011*; *St John et al., 2020a*; *Martin et al., 2019*; *St John et al., 2020b*). Furthermore, species boundaries persist across multiple lake populations, despite persistent admixture among species (*Martin and Feinstein, 2014*; *Richards et al., 2021*). We previously estimated fitness landscapes in these hypersaline lakes from two independent field experiments measuring the growth and survival of hybrids placed in field enclosures (*Figure 1b*). Selection analyses revealed a multi-peaked phenotypic fitness landscape that is stable across lake populations, year of study, and manipulation of the frequency of rare hybrid phenotypes (*Martin and Wainwright, 2013a*; *Martin and Gould, 2020*; *Martin, 2016*). One of the strongest and most persistent trends across studies and treatments was that hybrid phenotypes resembling the scale-eater were isolated in the lowest fitness region for both growth and survival relative to the other two species (*Martin and Wainwright, 2013a*; *Martin and Gould, 2020*). In contrast, hybrids resembling the generalist occupied a fitness peak and were separated by a smaller fitness valley from hybrids resembling the molluscivore, which occurred on a second peak of higher fitness.

Evolutionary trajectories through regions of low fitness should be inaccessible to natural selection. How then did an ancestral generalist population cross these phenotypic fitness valleys to reach new fitness peaks and adapt to novel ecological niches? A growing theoretical and empirical literature on

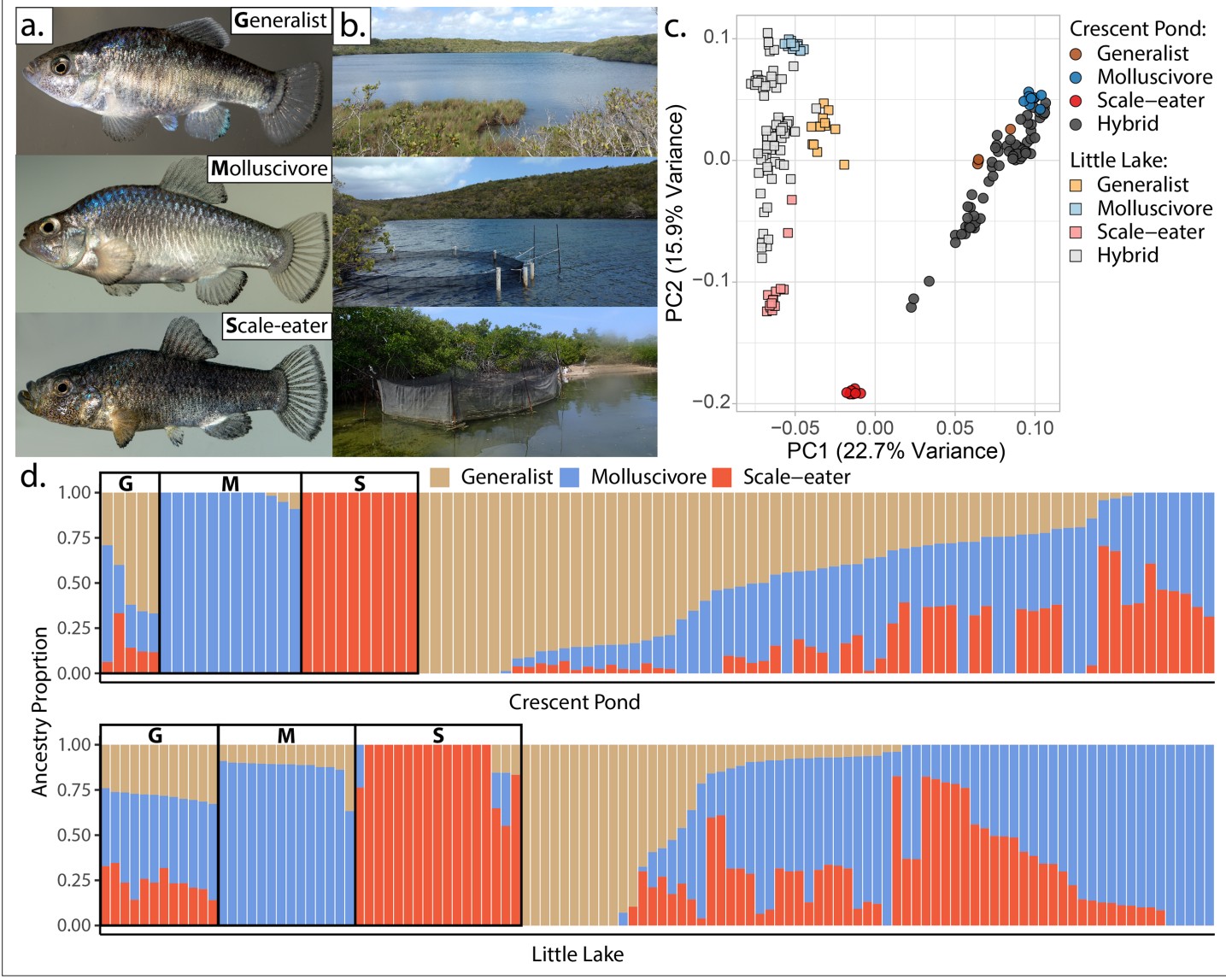

**Figure 1.** San Salvador Island pupfishes and their hybrids.

(**a**) From top to bottom: the generalist, *Cyprinodon variegatus*, the molluscivore *Cyprinodon brontotheroides*, and the scale-eater *Cyprinodon desquamator*. (**b**) Representative images of experimental field enclosures. (**c**) Principal component analysis of 1,129,771 linear discriminant (LD)-pruned single nucleotide polymorphisms (SNPs) genotyped in hybrids and the three parental species. (**d**) Unsupervised ADMIXTURE analyses for Crescent Pond (top) and Little Lake (bottom). G, M, and S indicate individual samples of generalists (G), molluscivores (M), and scale-eaters (S), respectively, followed by all resequenced hybrid individuals from field experiments. Colors indicate ancestry proportions in each population (K = 3).

The online version of this article includes the following figure supplement(s) for figure 1:

**Figure supplement 1.** Proportion (%) genetic variance explained by the first 20 principal components obtained using all single nucleotide polymorphisms (SNPs) and individuals from Crescent Pond, Little Lake, and Osprey Lake, as well as experimental hybrids.

**Figure supplement 2.** Principal components 2, 3, and 4.

**Figure supplement 3.** Supervised ADMIXTURE analyses for Crescent Pond (top) and Little Lake (bottom).

**Figure supplement 4.** Genetic distance predicts morphological distance among sampled hybrids.

**Figure supplement 5.** The proportion of generalist or specialist ancestry in hybrids did not predict fitness in experimental hybrids using either (a) composite fitness (tobit/zero-censored), (b) survival (binomial), or (c) growth (Gaussian).

fitness landscapes has demonstrated the limited conditions for crossing fitness valleys (*Weissman et al., 2010*; *Weissman et al., 2009*; *Iwasa et al., 2004*; *Bitbol and Schwab, 2014*). Fitness peaks and valleys in morphospace may result only from the reduction of the adaptive landscape to two phenotypic dimensions (*Wagner, 2012*). Additional phenotypic and genotypic dimensions may reveal fitness ridges that entirely circumvent fitness valleys (*Martin, 2016*; *Conrad, 1990*; *Whibley et al., 2006*). Indeed, owing to nonlinearity in the association between phenotype and fitness (*Martin et al., 2007*; *Gros et al., 2009*), even a single-peaked phenotypic fitness landscape may be underlaid by a multi-peaked genotypic fitness landscape (*Hwang et al., 2017*; *Park et al., 2020*). In this respect, investigating the high-dimensional genotypic fitness landscape is key to understanding the origins of novelty in this system, particularly given the rare evolution of lepidophagy (scale-eating), a niche occupied by less than 0.3% of all fishes (*Froese and Pauly, 2021*).

Furthermore, the relative contributions of standing genetic variation, de novo mutations, and adaptive introgression to the tempo and mode of evolution are now of central interest to the field of speciation genomics (*Seehausen, 2014*; *Martin and Jiggins, 2017*; *Marques et al., 2019*; *Nelson and Cresko, 2018*; *Thompson et al., 2019*). The three-dimensional adaptive landscape metaphor is often invoked to explain how the genetic, phenotypic, and ecological diversity introduced to populations by hybridization facilitates the colonization of neighboring fitness peaks that are unoccupied by either hybridizing species (*Mallet, 2007*; *Seehausen, 2004*; *Pardo-Diaz et al., 2012*). However, extension of these ideas to more high-dimensional genotypic fitness landscapes remains underexplored. For instance, we have yet to learn how the appearance of novel adaptive genetic variation through introgressive hybridization or de novo mutation alters the realized epistatic interactions among loci, thus potentially altering the shape of the fitness landscape and the accessibility of interspecific adaptive walks.

The adaptive radiation of San Salvador Island pupfishes, like many others (*Pease et al., 2016*; *Richards et al., 2018*; *Meier et al., 2017*; *McGee et al., 2020*; *Irisarri et al., 2018*), appears to have originated from a complex interplay of abundant standing genetic variation, adaptive introgression from neighboring islands, and several de novo single-nucleotide mutations and deletions found only in the scale-eater (*Richards et al., 2021*; *Richards et al., 2017*). Notably, both specialists harbor numerous introgressed single nucleotide polymorphisms (SNPs) showing evidence of hard selective sweeps in the regulatory regions of known craniofacial genes (*Richards et al., 2021*; *Richards et al., 2017*). In contrast, hard selective sweeps of de novo mutations only appear in the scale-eating species, *C. desquamator*. Here, we leverage whole-genome sequencing of 139 hybrids measured in field experiments to identify the genomic basis of fitness differences, infer genotypic fitness networks, summarize their topography, and quantify the accessibility of novel fitness peaks and the influence of each source of genetic variation on interspecific adaptive walks.

## Results

### Sample collection and genomic resequencing

We resequenced 139 hybrids (86 survivors, 56 deaths; *Supplementary file 1—table 1*) from two independent field experiments across a total of six field enclosures and two lake populations 2011: two high-density 3 m diameter enclosures exposed for 3 months: Crescent Pond n = 796; Little Lake n = 875 F2 hybrids (*Martin and Wainwright, 2013a*); 2014/2015: four high-density 4 m diameter enclosures exposed for 3 months in Crescent Pond, n = 923 F4/F5 hybrids and 11 months in Little Lake, n = 842 F4/F5 hybrids (*Martin and Gould, 2020*). We then characterized patterns of genetic variation among parental species in each lake and their lab-reared hybrids used in field experiments. We genotyped 1,129,771 SNPs with an average coverage of 9.79× per individual.

### Population structure and ancestry associations with fitness

Principal components analysis (PCA) of genetic variation strongly differentiated pupfishes sampled from Little Lake/Osprey Lake and Crescent Pond (PC1: 22.7% variance explained) and among species within each lake (PC2: 15.9% variance explained: *Figure 1d*; *Figure 1—figure supplements 1–2*). These results were supported by ADMIXTURE analyses (*Alexander et al., 2009*; *Alexander and Lange, 2011*; *Figure 1e*). However, some hybrids were genotypically transgressive, falling outside the genotypic distributions of the three parental species (*Figure 1—figure supplement 2*), leading

ADMIXTURE to assign the third cluster to these hybrids, rather than generalists which often contain segregating variation found in trophic specialists (*Froese and Pauly, 2021*). This pattern persisted in a supervised ADMIXTURE analysis, in which we assigned individuals from the three parental species a priori to their own population and estimated admixture proportions for the remaining hybrids (*Figure 1—figure supplement 3*). Pairwise genetic distances were significantly associated with pairwise morphological distances (*Figure 1—figure supplement 4*).

We analyzed three measures of fitness (growth, survival, and their composite: see Materials and methods and Supplement for details), but focus herein on composite fitness, which is equal to growth for survivors and zero for non-survivors. Growth could not be measured for tagged hybrids that died in field enclosures and thus were not recovered. Because reproductive success was not possible to quantify in field experiments (due to continuous egg-laying and very small, newly hatched fry), composite fitness included only measurements of growth and survival.

Interestingly, in no case were genome-wide patterns of parental ancestry in hybrids (estimated from unsupervised ADMIXTURE analyses) associated with hybrid composite fitness (generalist $p = 0.385$; scale-eater $p = 0.439$; molluscivore $p = 0.195$), growth (generalist $p = 0.119$; scale-eater $p = 0.283$; molluscivore $p = 0.328$), or survival probability (generalist $p = 0.440$; scale-eater $p = 0.804$; molluscivore $p = 0.313$) while controlling for effects of lake and experiment (*Figure 1—figure supplement 5*; *Supplementary file 1–table 2*). Similar results were obtained when repeating these analyses using admixture proportions estimated from a supervised ADMIXTURE analysis (*Supplementary file 1–table 3*), using only samples from the second field experiment (*Supplementary file 1–table 4*), or using principal component axes estimated from genome-wide SNPs (*Supplementary file 1–table 5*: see Supplementary results). Therefore, in contrast to previous studies (*Arnegard et al., 2014*; *Wang et al., 2002*; *Leimu et al., 2006*), in this system genome-wide ancestry is not consistently associated with fitness, highlighting the complex nonlinear relationship between genotype, phenotype, and fitness within this nascent adaptive radiation. We must look to local ancestry to understand fitness relationships (e.g. *Schluter et al., 2021*).

## Genome-wide association mapping of fitness

From our linear discriminant (LD)-pruned dataset, we used a linear mixed model (LMM) in GEMMA to identify 132 SNPs in regions that were strongly associated with composite fitness, including 13 which remained significant at the conservative Bonferroni-corrected threshold (*Figure 2a*, *Supplementary file 1–tables 6–7*; see supplement for results for survival and growth alone (*Supplementary file 1–tables 8–9*; *Figure 2—figure supplement 1*)). Gene ontologies for these 132 fitness-associated regions were significantly enriched for synaptic signaling and chemical synaptic transmission (false discovery rate [FDR] rate <0.01; *Figure 2—figure supplement 2*; *Supplementary file 1–table 7*). Ontologies enriched at an FDR rate <0.05 were related to signaling and regulation of cell communication (for growth, see *Figure 2—figure supplement 3*). We did not identify any enrichment for ontologies related to craniofacial development which have previously been identified to play a significant role in the adaptive divergence of these fishes (*Richards et al., 2021*; *Richards et al., 2017*; *McGirr and Martin, 2020*). This suggests that fitness-associated regions in our field experiments captured additional components of fitness beyond the external morphological phenotypes measured in previous studies.

We characterized whether genes in or near fitness-associated regions were implicated in adaptive divergence of the specialists. Surprisingly, no fitness-associated regions overlapped with regions showing significant evidence of a hard selective sweep (*Richards et al., 2021*). However, six fitness-associated genes were previously shown to contain either fixed divergent SNPs (*csad, glcci1, ino80c, mag, pim2, mettl21e*) or a fixed deletion between specialists (*med25*) (*McGirr and Martin, 2020*). *Med25* (mediator complex subunit 25) is a craniofacial transcription factor associated with cleft palate in humans and zebrafish (*Nakamura et al., 2011*; *Mork and Crump, 2015*); a precursor of *mag* (myelin-associated glycoprotein) was also associated with the parallel evolution of the thick-lipped phenotype in Midas cichlids based on differential expression among morphs (*Manousaki et al., 2013*). Three of the six remaining fitness-associated genes containing divergent SNPs (*McGirr and Martin, 2020*) were associated with growth and/or body size measurements in other fishes. First, *csad* plays an important role in synthesizing taurine which is a rate-limiting enzyme affecting growth rate in parrotfishes (*Lim et al., 2013*), rainbow trout (*Gaylord et al., 2006*), and Japanese flounder (*Yokoyama*

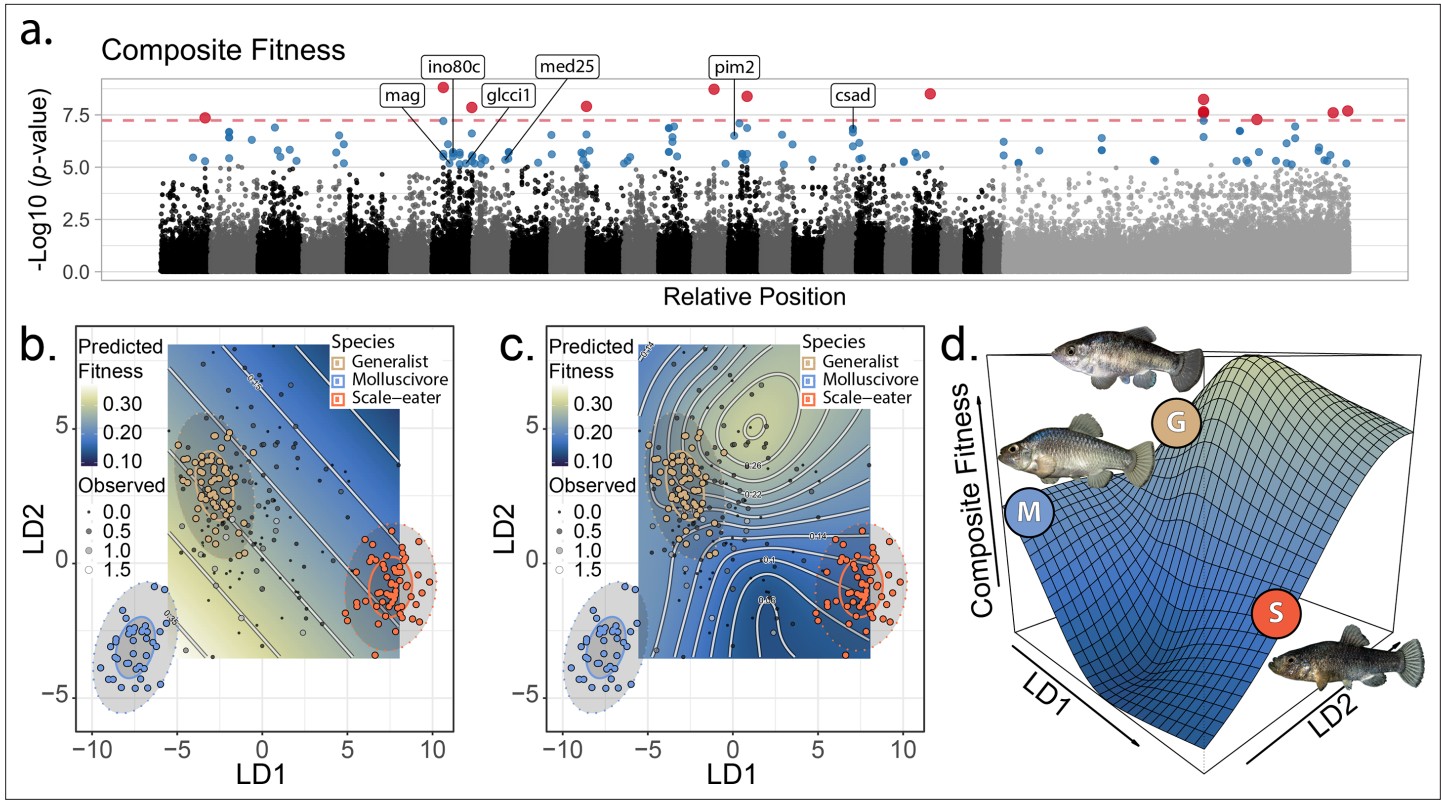

**Figure 2.** The genetic basis of fitness variation and improved inference of adaptive landscapes. (**a**) Per-single nucleotide polymorphism (SNP) log$_{10}$ $p$-values from a genome-wide association test with GEMMA for composite fitness (survival × growth). Lake and experiment were included as covariates in the linear mixed model. SNPs that were significant at false discovery rate (FDR) < 0.05 are indicated in blue; red SNPs above dashed red line cross the threshold for Bonferroni significance at α = 0.05. The first 24 scaffolds are sorted from largest to smallest and remaining scaffolds were pooled. The six genes associated with composite fitness which were both strongly differentiated (F$_{ST}$ > 0.95) and differentially expressed between specialists (**McGirr and Martin, 2020**) are annotated. (**b–c**) Best-fit adaptive landscape for composite fitness using either morphology alone (**b** flat surface with only directional selection) or morphology in combination with fitness-associated SNPs (**c** highly nonlinear surface). Best-fit model in **c** was a generalized additive model (GAM) including a thin-plate spline for both linear discriminant (LD) axes, fixed effects of experiment and lake, and fixed effects of the seven (see Supplementary methods) SNPs most strongly associated with fitness shown in red in panel a. (**d**) Three-dimensional view of **c** with relative positions of the three parental phenotypes indicated.

The online version of this article includes the following figure supplement(s) for figure 2:

**Figure supplement 1.** Manhattan plots illustrating the strength of association between individual single nucleotide polymorphisms (SNPs) and either survival (**A**) or growth (**B**) as inferred by GEMMA.

**Figure supplement 2.** Gene ontology enrichment for single nucleotide polymorphisms (SNPs) found to be associated with composite fitness.

**Figure supplement 3.** Gene ontology enrichment for single nucleotide polymorphisms (SNPs) found to be associated with growth.

**Figure supplement 4.** The 29 landmarks used to digitally measure 30 linear traits plus standard length using DLTDV8a (**Hedrick, 2008**).

**Figure supplement 5.** Morphological variation in the three San Salvador Island pupfish species and their experimentally produced hybrids.

**Figure supplement 6.** Best-fit fitness landscapes for composite fitness (**a**) survival (**b**), growth without associated single nucleotide polymorphisms (SNPs) (**c**), and growth including associated SNPs (**d**).

**Figure supplement 7.** Comparison of 10,000 bootstrapped estimates of predicted mean composite fitness to estimations from observed data across slices of the fitness landscape.

**Figure supplement 8.** The topography of the composite fitness adaptive landscape is influenced by the distribution of a common single nucleotide polymorphism (SNP) haplotype.

et al., 2001). Second, *glcci1* is associated with the body depth/length ratio in yellow croaker (**Zhou et al., 2019**). Third, *ino80c* is associated with measures of body size in Nile tilapia (**Yoshida and Yáñez, 2021**). Finally, *mettl21e* was differentially expressed among specialists and also misexpressed in F1 hybrids between scale-eaters and molluscivores at 8 days post-fertilization and thus is a putative

genetic incompatibility in this system that may impact their fitness in field enclosures (*McGirr and Martin, 2020*; *Kulmuni and Westram, 2017*). Although it has not been associated with growth or body size in fishes, *mettl21e* is associated with intramuscular fat deposition in cattle (*Fonseca et al., 2020*). Taken together, these findings support the interpretation that fitness-associated regions are associated with unmeasured traits, particularly physiological growth rate, or craniofacial shape in the case of the deletion in *med25*, that affect fitness in our hybrid field experiments. However, the fitness-associated loci we identified appear not to have the subject of selective sweeps in either specialist.

## Fitness-associated SNPs improve inference of the adaptive landscape

Fitness landscapes in past studies were estimated using slightly different sets of morphological traits; thus, to enable inclusion of all hybrids on a single fitness landscape, a single observer (AHP) remeasured all sequenced hybrids for 31 morphological traits (*Figure 2—figure supplement 4*; *Supplementary file 1–tables 1–10*). We used linear discriminant axes and generalized additive models (GAM) to estimate phenotypic fitness landscapes for the sequenced hybrids on a two-dimensional morphospace indicating similarity to each of the three parental populations following previous studies (*Martin and Wainwright, 2013a*; *Martin and Gould, 2020*; *Figure 2—figure supplement 5*; *Supplementary file 1–tables 11–13*). We then tested whether inclusion in the GAM of the 13 genomic regions most strongly associated with fitness (red: *Figure 2a*) improved our inference of the underlying adaptive landscape. Models including fitness-associated SNPs were invariably favored over models with external morphology alone (ΔAICc > 8.6: *Supplementary file 1–tables 14–15*). Morphology-only models predicted a flat fitness surface (*Figure 2b*, *Figure 2—figure supplement 6*; predictions restricted to observed hybrid morphospace). In contrast, models including fitness-associated SNPs predicted a complex and nonlinear fitness landscape, despite our limited dataset of 139 sequenced hybrids relative to samples in previous morphology-only studies of >800 hybrids per enclosure.

To reduce complexity of the full model estimated from 31 morphological traits including all 13 fitness-associated SNPs, we fit an additional model including only the seven most significant fitness-associated SNPs in the full model. This reduced model was the best fit; the inferred adaptive landscape was complex and characterized by a fitness peak near hybrids resembling the generalist phenotype separated by a small fitness valley from a second region of high fitness for hybrids resembling the molluscivore phenotype. Hybrids resembling the scale-eater phenotype again occurred in a large fitness valley (*Figure 2b–2d*: For results pertaining to growth or survival, see *Supplementary file 1*: *Figure 2—figure supplement 6*, *Supplementary file 1–tables 11–15*). Each of these fitness peaks and valleys were frequently recovered across 10,000 bootstrap replicates; landscapes inferred from bootstrap replicates were often more complex with increased curvature relative to inferences from our observed dataset (*Figure 2—figure supplement 7*). Thus, the fitness landscape estimated from our observed dataset appears robust to sampling uncertainty.

Compared to previous studies, the highest fitness optimum was shifted from the molluscivore to the generalist phenotype. This suggests that fitness-associated SNPs increased the fitness of hybrids resembling generalists beyond expectations based on their morphology alone, consistent with the hypothesis that fitness-associated SNPs are associated with unmeasured non-morphological traits affecting fitness. Indeed, visualization of observed haplotypes in hybrids across the fitness landscape supported this interpretation; one of the most common haplotypes was most frequent in hybrids resembling generalists near the peak of high fitness and rare in hybrids resembling either trophic specialist (*Figure 2—figure supplement 8*). Regardless, this two-dimensional phenotypic fitness landscape did not reveal fitness ridges connecting generalists to specialists, further emphasizing the need to investigate the genotypic fitness landscape.

## Trophic novelty is associated with isolation on the genotypic fitness network

The adaptive radiation of pupfishes on San Salvador Island originated within the last 10,000 years through a combination of selection on standing genetic variation, adaptive introgression, and de novo mutations (*Richards et al., 2021*). However, it is unclear how each source of genetic variation aided in the traversal of fitness paths or contributed to the colonization of novel fitness peaks. To address this knowledge gap, we first sought to visualize genotypic fitness networks and gain insight into how isolated the three species are in genotypic space. Understanding the relative isolation of each

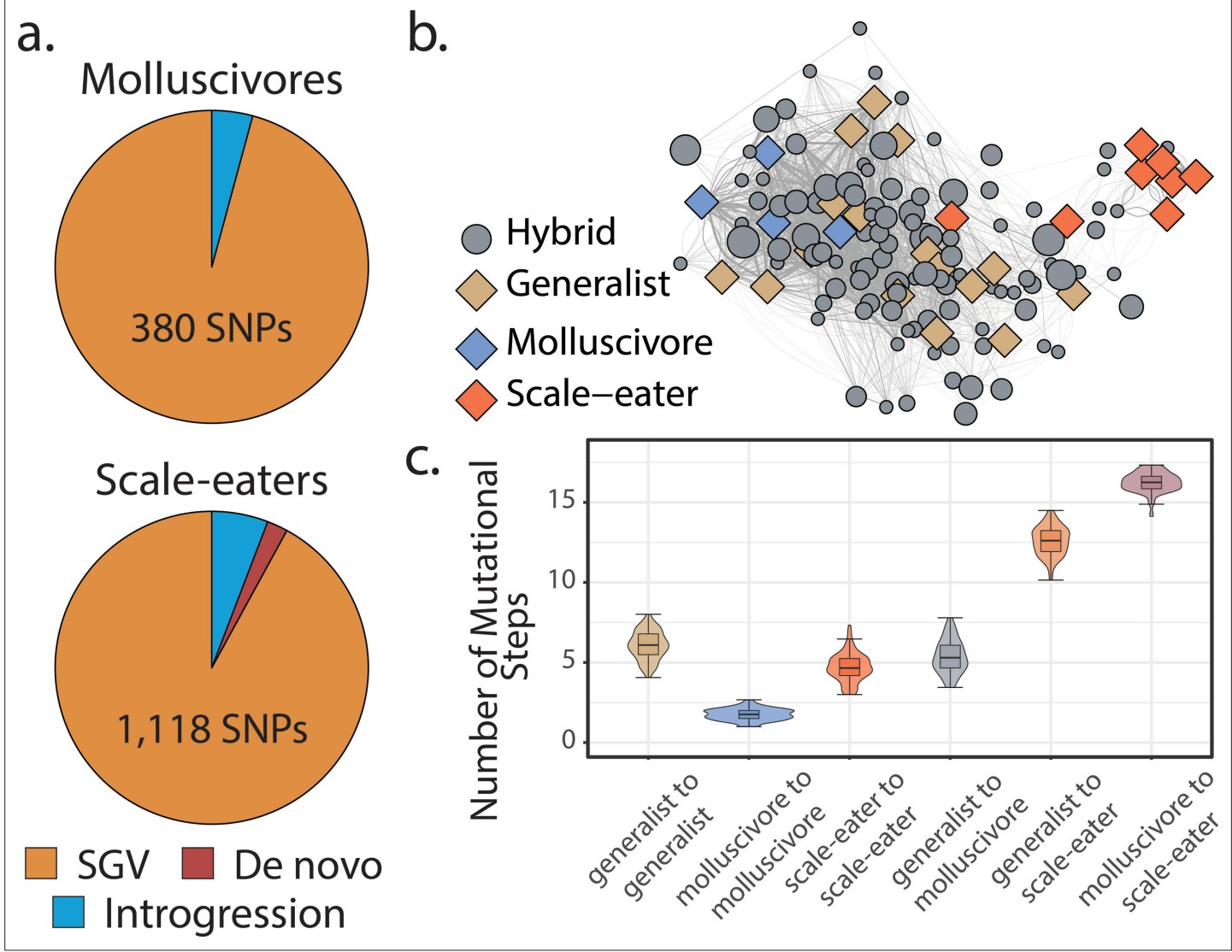

**Figure 3.** Scale-eaters are isolated on the fitness landscape.

(a) Most nearly fixed or fixed variants ($F_{ST} \geq 0.95$) experiencing hard selective sweeps (hereafter 'adaptive alleles') originated as standing genetic variation (SGV: molluscivores = 96%, scale-eaters = 92%), followed by introgression (molluscivores = 4%, scale-eaters = 6%), and de novo mutation (scale-eaters = 2%)(*Richards et al., 2021*). Pie charts show adaptive alleles retained in our study for each species; networks are constructed from either set of adaptive alleles. (b) Genotypic network constructed from a random sample of 10 single nucleotide polymorphisms (SNPs), sampled from all SNPs shown in **a**. Each edge between nodes is up five mutational steps away; edge width is proportional to mutational distance: wider edges connect closer haplotypes; hybrid node size is proportional to fitness (larger nodes are of greater fitness value). (c) Median number of mutational steps within or between species (e.g. *Figure 4a*). All pairwise comparisons using Tukey's HSD test (after false discovery rate [FDR] correction) were significant.

specialist from the generalist can reveal the relative accessibility of their respective adaptive walks on the genotypic fitness landscape.

To accomplish this, we reconstructed genotypic fitness networks from 1498 candidate adaptive alleles previously identified in this system (e.g. *Figure 3a*; *Richards et al., 2021*). These regions displayed significant evidence of a hard selective sweep using both site frequency spectrum and LD-based methods, SweeD (*Pavlidis et al., 2013*) and OmegaPlus (*Alachiotis et al., 2012*), and contained fixed or nearly fixed SNPs ($F_{ST} > 0.95$) differentiating trophic specialists across lakes (*Richards et al., 2021*). Adaptive alleles were classified as standing variation, introgressed, or de novo mutations based on extensive sampling of focal and related *Cyprinodon* pupfish species across San Salvador Island and neighboring Caribbean islands, as well as North and South American outgroups (*Richards et al., 2021*). We note, however, that adaptive alleles designated as de novo on San Salvador

Island may be segregating at low frequencies in other sampled populations or present in unsampled populations.

These fitness networks depict both hybrids and parental species in genotypic space, with nodes representing SNP haplotypes and edges connecting mutational neighbors (*Figure 3b*). Genotypic space is immense; using SNPs coded as homozygous reference, heterozygote, or homozygous alternate, the number of potential haplotypes is equal to $3^{\text{\# SNPs in network}}$. For instance, to construct a reduced network of 100 SNPs, there are a total of $3^{100} = 5.17 \times 10^{57}$ possible nodes. Thus, unlike experimental studies of individual proteins in haploid *E. coli* (*Weinreich et al., 2006*; *Khan et al., 2011*) or yeast (*Bank et al., 2016*), it is not possible for us to investigate the full breadth of genotypic space.

Instead, to understand the distribution of parental species and their hybrids in genotypic space, we began by using a random sample of 10 SNPs drawn from our set of candidate adaptive alleles in this system. Here, we plotted edges between nodes up to five mutational steps away (e.g. *Figure 3b*) and found that generalists and molluscivores are closer on the genotypic fitness network than either is to scale-eaters (*Figure 3c*), as expected based on their genetic distance. Most scale-eaters appear quite isolated in genotypic space, separated from the generalist cluster of nodes by 12.6 ± 0.091 (mean ± SE: $p < 0.001$) mutational steps and from molluscivores by 16.3 ± 0.060 steps ($p < 0.001$). In contrast, molluscivores were separated from generalists by 5.37 ± 0.103 steps ($p < 0.001$). Generalists show the greatest intrapopulation distances, separated from each other by 6.08 ± 0.088 steps ($p < 0.001$). In contrast, molluscivores exhibited the smallest intrapopulation distances, separated by 1.75 ± 0.021 steps ($p < 0.001$). Scale-eater intrapopulation distances were intermediate (4.71 ± 0.088 steps: $p < 0.001$).

## Molluscivore genotypes are more accessible to generalists than scale-eater genotypes on the genotypic fitness landscape

The most accessible paths through genotypic fitness networks are characterized by monotonically increasing fitness at each mutational step and the smallest possible number of steps between two states (*Fragata et al., 2019*; *Weinreich et al., 2006*; *Franke et al., 2011*; *Figure 4a–b*). Furthermore, as described earlier, the accessibility of individual fitness peaks is predicted to be reduced on increasingly rugged fitness landscapes that are characterized by a greater number of fitness peaks (*Kauffman and Levin, 1987*; *Neidhart et al., 2014*; *Franke et al., 2011*; *de Visser and Krug, 2014*). This provides three useful metrics of evolutionary accessibility for genotypic trajectories: (1) the total number of accessible paths relative to network size (*Figure 4—figure supplement 1*; *Supplementary file 1–table 16*), (2) the length of the shortest accessible paths, and (3) the number of fitness peaks (ruggedness). Here, we define peaks as genotypes with no fitter neighbors and within a single mutational step (*Ferretti et al., 2018*). With these three metrics, we can quantify the accessibility of interspecific genotypic pathways.

We used these measures of accessibility to ask: (1) whether molluscivore or scale-eater genotypes were more accessible to generalists on the fitness landscape (*Figure 4c–d*) and (2) whether molluscivore and scale-eater genotypic fitness networks differed in their ruggedness, characterized by peak number (*Figure 4e–g*). These measures provide insight into the predictability of evolution and the role that epistasis plays in their evolution (*Kauffman and Levin, 1987*; *Wright, 1931*; *Neidhart et al., 2014*; *Whitlock et al., 1995*).

We constructed 5000 genotypic fitness networks from a random sample of five species-specific candidate adaptive SNPs (*Figure 3a*) for either molluscivores or scale-eaters, requiring that at least one SNP of each source of genetic variation be present in the sample. We used odds ratios (ORs) to compare the relative accessibility and ruggedness of molluscivore fitness networks compared to scale-eater networks (*Figure 4h*). Thus, ORs greater than 1 imply summary statistics are greater for molluscivores than for scale-eaters.

We found that molluscivore genotypes were significantly more accessible to generalists on the fitness landscape than scale-eaters (*Supplementary file 1–table 17*); molluscivore networks had significantly more accessible paths [OR: (95% CI) = 2.095: (1.934, 2.274)] that were significantly shorter [OR and 95% CI = 0.253: (0.231, 0.277)]. Not only were molluscivore genotypes more accessible to generalists, but molluscivore fitness networks were significantly less rugged than scale-eater networks, comprised of fewer peaks [OR and 95% CI = 0.604: (0.575, 0.634)], and connected by significantly

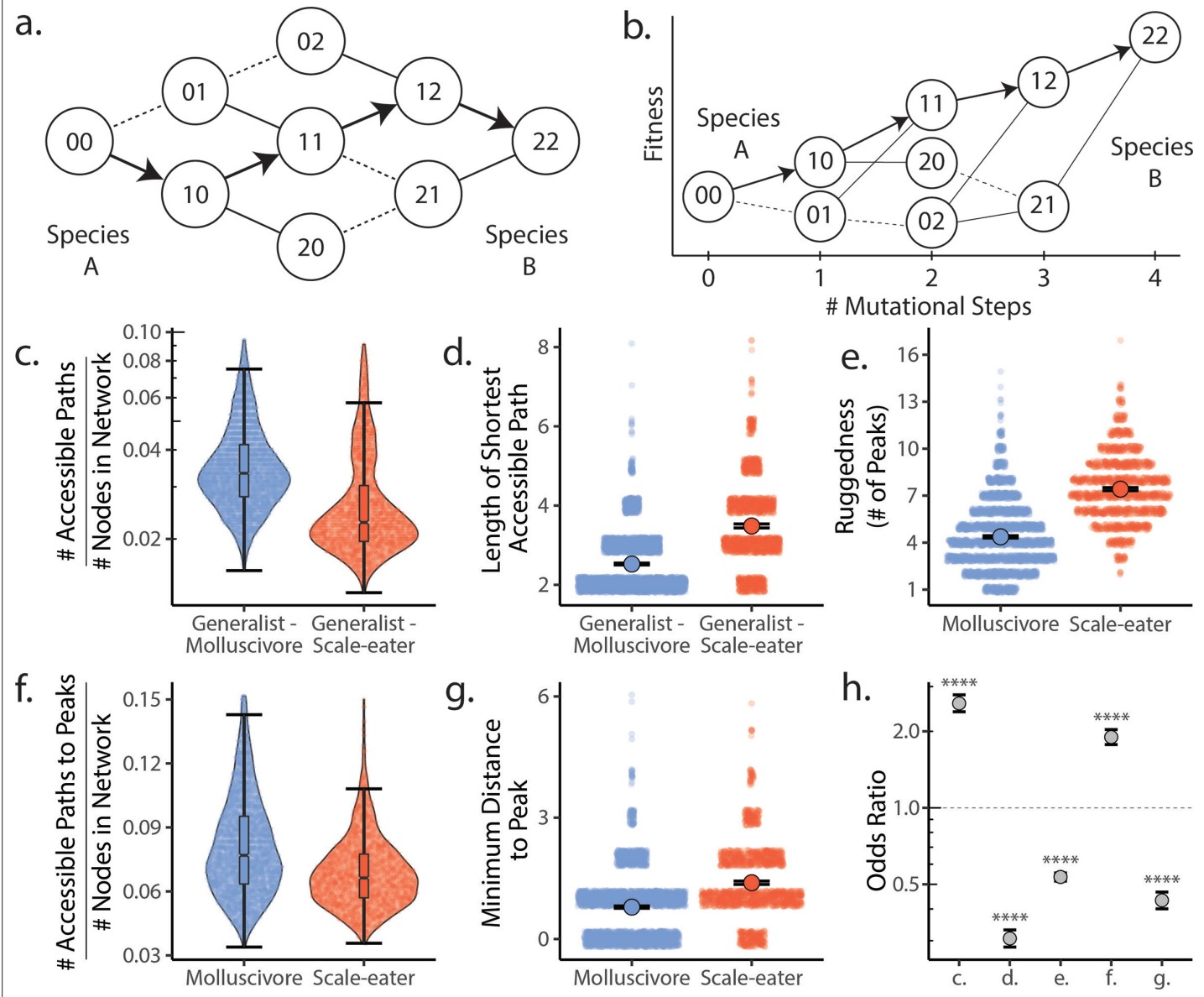

**Figure 4.** Molluscivore genotypes were more accessible to generalists on the genotypic fitness landscape than scale-eater genotypes.
(**a**) Diagram illustrating genotypic fitness networks and adaptive walks between species for a hypothetical two-single nucleotide polymorphism (SNP) genotypic fitness landscape. Species A and B are separated by four mutational steps. Dashed lines indicate inaccessible paths that decrease in fitness leaving a single possible accessible evolutionary trajectory between species A and B (indicated by bold arrows). Each node in our study is associated with an empirical measure of hybrid fitness from field experiments (**Martin and Wainwright, 2013a**; **Martin and Gould, 2020**). Edges are always drawn as directed from low to high fitness. (**b**) The same network as in (**a**), with fitness plotted on the y-axis and number of mutational steps from species A to B on the x-axis. The only accessible path between species A and B is indicated by solid arrows. (**c**) Number of accessible paths between generalists and either specialist, scaled by network size. (**d**) Length (# of nodes) of the shortest accessible paths. Means (large points) ± 2 standard errors are plotted. (**e**) Ruggedness, as measured by the number of peaks (genotypes with no fitter neighbors within a single mutational step; **Ferretti et al., 2018**). (**f**) Number of accessible paths to peaks, scaled by network size. (**g**) Length of the shortest accessible path to the nearest peak. (**h**) Odds ratios (OR: maximum likelihood estimate and 95% CI) for each measure of accessibility (x-axis corresponds to panel letters); molluscivore networks have significantly greater summary statistics when OR > 1. Molluscivore genotypes are more accessible to generalists than scale-eater genotypes due to a significantly greater number of accessible paths separating them (**c**) that are significantly shorter (**d**). Molluscivore genotypic networks were also less rugged, that is, they contained significantly fewer peaks (**e**), each of which were in turn more accessible from the generalist genotypes (**f**, **g**).

The online version of this article includes the following figure supplement(s) for figure 4:

**Figure supplement 1.** The raw number of accessible paths increases with network size.

more accessible paths [OR and 95% CI = 1.514: (1.404, 1.635)], that contained fewer mutational steps [OR and 95% CI = 0.539: (0.500, 0.579)].

## Adaptive introgression and de novo mutations increase accessibility of novel fitness peaks

We further used our two metrics of accessibility and landscape ruggedness to ask how different sources of adaptive genetic variation may influence the topography of the fitness landscape, the traversal of fitness paths separating generalists from specialists and ultimately colonization of novel fitness peaks. We constructed genotypic fitness networks limited to only one of the three main sources of adaptive genetic variation: standing genetic variation, introgression from one of four focal Caribbean generalist populations, or de novo mutations unique to San Salvador Island. We also examined all combinations of these three sources to better reflect the actual process of adaptive divergence originating from only standing genetic variation, then adaptive introgression plus standing genetic variation, and finally the refinement stage of de novo mutations (*Richards et al., 2021*).

We compared sets of 5000 random five-SNP genotypic networks drawn from different sources of adaptive variation (*Figure 4a*) and compared the effect of each source of variation on measures of accessibility and landscape ruggedness relative to standing genetic variation. We treated standing variation as our basis for comparison because this is the source of genetic variation first available to natural selection (*Barrett and Schluter, 2008*).

We discovered that genotypic trajectories between generalists and either trophic specialist in genotypic fitness networks constructed from introgressed or de novo adaptive mutations were significantly more accessible than networks constructed from standing genetic variation (*Figure 5*). Specifically, random networks that included alternate sources of adaptive variation contained significantly more accessible fitness paths from generalist to specialists than networks constructed from standing genetic variation alone, while controlling for differences in overall network size (*Figure 5a*; *Supplementary file 1—table 18*). Furthermore, accessible paths between generalists and specialists in networks constructed from introgressed or de novo adaptive loci were significantly shorter in length (*Figure 5b*). We recovered the same pattern whether constructing fitness networks from these sources of variation alone or in combination. These results held across all measures of fitness and for analyses repeated using only hybrids sampled from the second field experiment (*Figure 5—figure supplements 1–2*, *Supplementary file 1—tables 18–19*).

Our finding of increased accessibility of interspecific genotypic trajectories suggests that fitness landscapes constructed from adaptive standing genetic variation alone are more rugged than networks including adaptive loci originating from either introgression or de novo mutation. Quantification of landscape ruggedness supported this hypothesis in all cases (*Figure 5c*; *Supplementary file 1—tables 18–19*). Additionally, increasing landscape ruggedness significantly decreased the length of accessible paths to the nearest local peak [glm(min. path length ~ number of peaks, family = poisson): $p < 0.0001$, β = –0.088, 95% CI = –0.095 to 0.081].

Scale-eater fitness genotypic fitness landscapes constructed from a combination of adaptive loci sourced from standing variation, introgression, and de novo mutations had significantly more accessible paths (scaled by network size) separating generalists from scale-eaters [OR and 95% CI = 1.879: (1.743, 2.041); LRT $p < 0.0001$; *Figure 5a*] and these paths were significantly shorter in length compared to networks constructed from standing variation alone [OR and 95% CI = 0.876: (0.823, 0.932); LRT $p < 0.0001$; *Figure 5b*]. The only exception to this pattern across all three fitness measures was for growth rate in genotypic fitness networks constructed for molluscivore adaptive loci; no significant difference was observed in the length of the shortest accessible path between networks constructed using standing variation alone or those constructed using introgressed alleles [OR and 95% CI = 0.994: (0.915, 1.079); LRT $p = 0.8826$; *Supplementary file 1—table 18*]. Interestingly, however, for networks constructed from standing variation and introgressed alleles, we again observed a significant reduction in length of the shortest accessible paths [OR and 95% CI = 0.897: (0.835, 0.962); LRT $p = 0.0050$; *Supplementary file 1—table 18*].

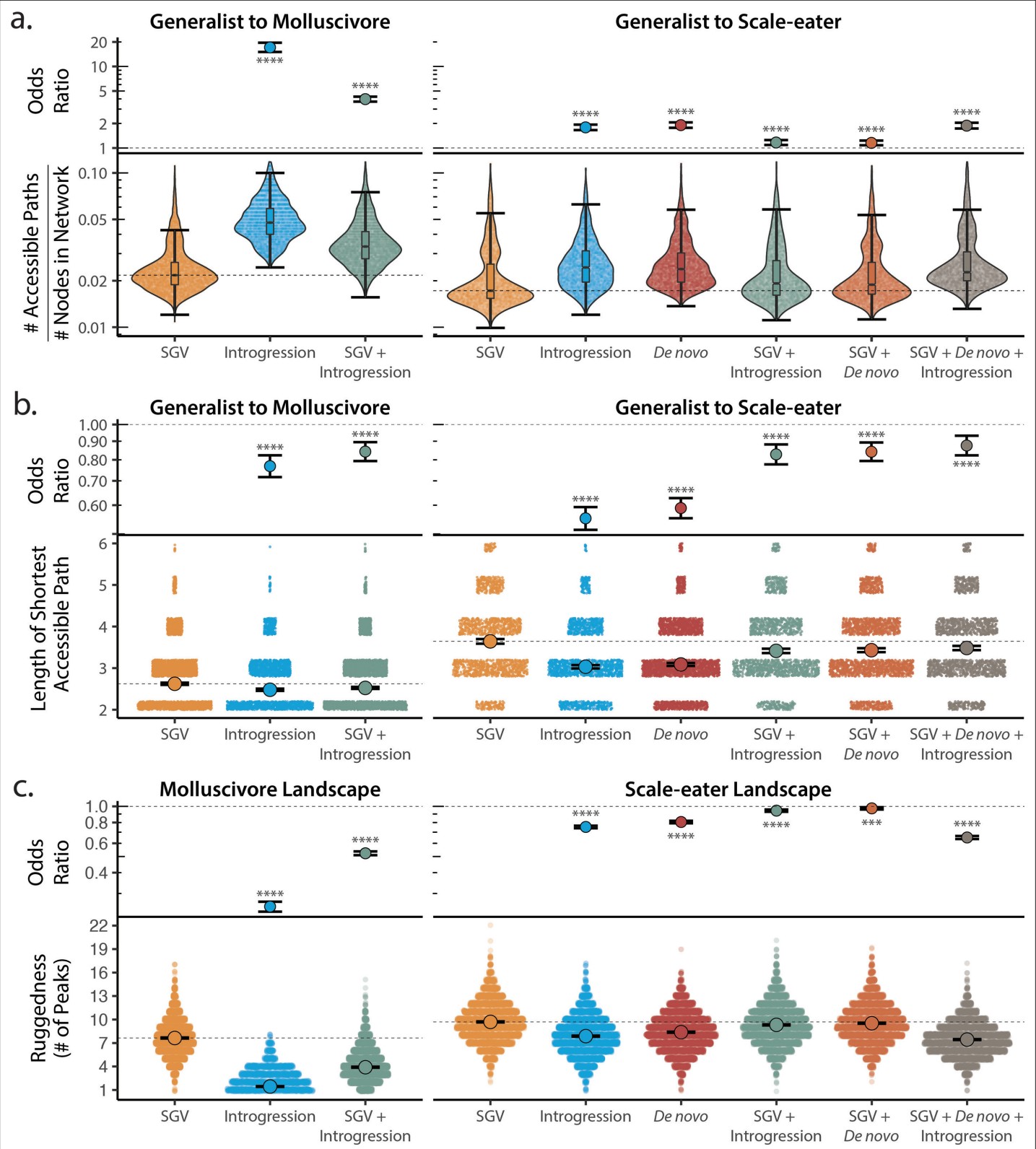

**Figure 5.** Adaptive introgression and de novo mutations increase access to specialist fitness peaks. Odds ratios (maximum likelihood estimate and 95% CI) indicate the effect of each source of variation on accessibility compared to networks estimated from standing variation alone. Asterisks denote significance ($p < 0.0001$ = ****, $< 0.001$ = ***). (**a**) The number of accessible (i.e. monotonically increasing in fitness) paths per network, scaled by the size of the network (# of nodes in network). Significance was assessed using a likelihood ratio test, corrected for the false discovery rate (reported in

*Figure 5 continued on next page*

*Figure 5 continued*

**Supplementary file 1–table 18**). Dashed lines correspond to the median estimate for standing genetic variation to aid comparison to other sources of adaptive variation. (**b**) Number of mutational steps in the shortest accessible path. Means are plotted as large circles, with two standard errors shown; dashed horizontal lines correspond to the mean for standing genetic variation. (**c**) Ruggedness of molluscivore and scale-eater genotypic fitness networks constructed from each source of genetic variation measured by the number of peaks (genotypes with no fitter neighbors).

The online version of this article includes the following figure supplement(s) for figure 5:

**Figure supplement 1.** Adaptive loci sourced from introgression and de novo mutation reduce fitness landscape ruggedness and increase accessibility as compared to standing genetic variation (SGV) using survival as our proxy for fitness.

**Figure supplement 2.** Adaptive loci sourced from introgression and de novo mutation reduce fitness landscape ruggedness and increase accessibility as compared to standing genetic variation (SGV) using growth as our proxy for fitness.

## Discussion

We developed a new approach for estimating genotypic fitness landscapes for diploid organisms and applied it to a system in which phenotypic fitness landscapes have been extensively investigated. We were able to address long-standing questions posed by fitness landscape theory in an empirical system and assess the extent to which the shape of the fitness landscape and accessibility of adaptive walks are contingent upon the source of adaptive genetic variation. We show that not only are scale-eaters more isolated than molluscivores from generalists on the fitness landscape, but that the scale-eater fitness landscape is more rugged than molluscivores. This indicates that epistasis is more pervasive on the scale-eater fitness landscape, leading to less predictable evolutionary outcomes and fewer accessible trajectories from generalist to scale-eater genotypes. Overall, we found that most genotypic trajectories were inaccessible and included one or more mutational steps that decreased in fitness from generalist to specialist. This finding is consistent with the patterns observed by *Weinreich et al., 2006*, who constructed combinatorially complete fitness networks for five mutations contributing to antibiotic resistance in *E. coli* and found that only 18 of 120 possible genotypic trajectories were evolutionarily accessible. In contrast, *Khan et al., 2011* estimated that over half of all trajectories were accessible on a complete fitness landscape constructed using the first five adaptive mutations to fix in an experimental population of *E. coli*.

We also show that fitness landscapes are most rugged, and therefore epistasis most pervasive, when constructed from standing genetic variation alone, ultimately leading to a reduction in the accessibility of fitness peaks on these landscapes (*Figure 5*). This finding has significant implications for the predictability of evolution in the earliest stages of the speciation process. Adaptation from standing genetic variation is thought to initially be more rapid due to its initial availability and potentially reduced genetic load within a population (*Barrett and Schluter, 2008*; *Hermisson et al., 2017*; *Hedrick, 2013*). In contrast, we consistently found that networks constructed from a combination of adaptive standing variation, introgression, and de novo mutations reduced the ruggedness of fitness landscapes and thus increased accessibility of interspecific evolutionary trajectories (*Figure 5*). This would suggest that adaptive introgression or de novo mutations reduce the impacts of epistasis, resulting in a smoother fitness landscape with a greater number of accessible adaptive walks, facilitating the colonization of new adaptive zones. Future studies testing the generality of these findings will be invaluable for our understanding of the speciation process.

Furthermore, our results shed light on the classic problem of crossing fitness valleys on three-dimensional phenotypic fitness landscapes. We show that phenotypic fitness valleys may be circumvented by rare accessible paths on the genotypic fitness landscape. These results are consistent with increasing recognition that three-dimensional depictions of the fitness landscape may lead to incorrect intuitions about how populations evolve (*Pigliucci and Müller, 2010*; *Fragata et al., 2019*; *Kaplan, 2008*).

Our study represents a significant contribution to the growing body of work applying fitness landscape theory to empirical systems (*Karageorgi et al., 2019*; *Nosil et al., 2020*; *Pokusaeva et al., 2019*; *Gong et al., 2013*; *Gong and Bloom, 2014*). Unlike previous studies that experimentally generated combinatorially complete fitness landscapes (*Bank et al., 2016*; *Weinreich et al., 2006*; *Khan et al., 2011*), we subsampled loci across the genome, enabling us to quantify aspects of the genotypic fitness landscape, despite the limitations imposed by large genome sizes and non-model vertebrates. One limitation of this approach is that subsampled fitness networks may not directly correspond

to the full landscape (*Fragata et al., 2019*; *Blanquart and Bataillon, 2016*). For instance, a given subsampled fitness landscape may be present on multiple global, fully sampled fitness landscapes (*Blanquart and Bataillon, 2016*). Second, nodes (here, SNP haplotypes) can appear disconnected in a subsampled fitness landscape, but may be connected in the full fitness landscape (*Fragata et al., 2019*). Nevertheless, given that there are more possible genotypes for a gene of 1000 base-pairs than particles in the known universe (*Wright, 1932*; *Szendro et al., 2013*), nearly all empirical fitness landscapes must necessarily be subsampled at some scale.

Although inferences from subsampled fitness networks have their limitations, so too do those obtained from combinatorially complete fitness landscapes, which may themselves be misleading (*Whitlock et al., 1995*). By including mutations that are not segregating in natural populations, the shape of the 'complete' fitness landscape and thus accessibility of fitness peaks may be quite different from what occurs in nature. The shape of fitness landscapes in nature is dictated by the 'realized' epistasis that occurs among naturally segregating loci (*Whitlock et al., 1995*). Changes to 'realized' epistasis induced by introgression or de novo mutations appear to be one mechanism altering the shape of the fitness landscape and thus accessibility of fitness peaks. Our findings that adaptive introgression and de novo mutations make fitness peaks more accessible points toward a pervasive role of epistasis in determining the predictability of evolution and the speciation process (*Fragata et al., 2019*; *Kauffman and Levin, 1987*; *Bank et al., 2016*; *Wright, 1931*; *Aita et al., 2001*; *Neidhart et al., 2014*).

In the present study we have taken snapshots of the fitness landscape from loci that have already undergone hard selective sweeps. Consequently, we cannot directly assess the influence of each adaptive allele on the fitness landscape through time as it increases in frequency. However, so far we have failed to detect evidence of frequency-dependent selection in this system after experimental manipulations, at least for morphological traits (*Martin and Gould, 2020*). Future experimental or simulation studies may track how novel adaptive alleles affect fitness landscape topography as they increase in frequency.

## Conclusion

Our findings are consistent with a growing body of evidence that de novo and introgressed adaptive variation may contribute to rapid speciation and evolution toward novel fitness peaks (*Blount et al., 2008*; *Marques et al., 2019*; *Meier et al., 2017*; *Nelson et al., 2021*; *Svardal et al., 2020*; *Lamichhaney et al., 2018*; *Grant and Grant, 2019*; *Edelman and Mallet, 2021*). We demonstrate that adaptive introgression smooths the fitness landscape and increases the accessibility of fitness peaks. This provides an alternative mechanism to explain why hybridization appears to play such a pervasive role in adaptive radiation and speciation. There are many examples of hybridization promoting or inducing rapid speciation and adaptive radiation. Whether in Galapagos finches (*Grant and Grant, 2019*), African cichlids (*Richards et al., 2018*; *Meier et al., 2017*; *Svardal et al., 2020*; *Meier et al., 2019*; *Poelstra et al., 2018*), or *Heliconius* butterflies (*Pardo-Diaz et al., 2012*; *Moest et al., 2020*), hybridization has been shown to play a generative role in adaptive radiation and the evolution of novelty. One mechanism is the increased genotypic, phenotypic, and ecological diversity generated by hybridization in the form of transgressive phenotypes (*Seehausen, 2004*; *Lewontin and Birch, 1966*; *Rieseberg et al., 1999*; *Kagawa and Takimoto, 2018*; *Abbott et al., 2013*). This diversity in turn facilitates the colonization of novel fitness peaks and ecological niches, particularly after colonization of a new environment rich in ecological opportunity (*Mallet, 2007*; *Seehausen, 2004*; *Edelman and Mallet, 2021*). However, this model often assumes that the fitness landscape remains static after adaptive introgression. Here, we show that adaptive introgression directly alters the shape of the fitness landscape, making novel fitness peaks more accessible to natural selection. Thus, hybridization not only generates genetic diversity, but this diversity can alter the shape of the fitness landscape, changing which genotypic combinations are favored by natural selection along with the adaptive walks that lead to them.

## Materials and methods

### Sampling

Our final genomic dataset was comprised of 139 hybrids subsampled from two separate field experiments (*Martin and Wainwright, 2013a*; *Martin and Gould, 2020*) on San Salvador Island. Experiments were conducted in two lakes: Little Lake (N = 71) and Crescent Pond (N = 68). Hybrids in the first field experiment (*Martin and Wainwright, 2013a*) were comprised of F2 and backcrossed outbred juveniles resulting from crosses between all three species. Juveniles were raised for 2 months in the lab, individually tagged by injecting a stainless steel sequential coded wire tag (Northwest Marine Technologies, Inc) into their left dorsal musculature, and photographed pre-release for morphometric analyses. Experimental field enclosures consisted of high- and low-density treatments; density was varied by the number of tagged juveniles released into each enclosure. Hybrids in the second field experiment (*Martin and Gould, 2020*) were comprised of F4–F5 outbred juveniles resulting from crosses between all three species. Individuals were spawned, raised, tagged, and photographed in the same way prior to release. The second field experiment consisted of high- and low-frequency treatments of approximately equal densities. The frequency of rare transgressive hybrid phenotypes was manipulated between treatments in each lake, such that the high- and low-frequency treatments harbored an artificially increased and decreased frequency of transgressive phenotypes, respectively (*Martin and Gould, 2020*).

All hybrids were measured for 32 external morphological traits (see below). Additionally, we sequenced parental species of the generalist (N = 17), molluscivores (N = 27), and scale-eaters (N = 25) sampled from these two lakes and previously included in *Richards et al., 2021*. Note that we treated samples from Little Lake and Osprey Lake as the same population because these two lakes are connected through a sand bar and fish from these populations are genetically undifferentiated (*Martin and Feinstein, 2014*; *Richards et al., 2021*). For morphological analyses, we additionally measured samples of 60 generalists, 38 molluscivores, and 60 scale-eaters raised in the same laboratory common garden environment as the hybrids used in field experiments. A full list of samples is included in the supplement (*Supplementary file 1–table 1*).

### Sequencing, genotyping, and filtering

Raw reads from a combined set of 396 samples (see *Supplementary file 1*) were first mapped to the *C. brontotheroides* reference genome (genome size = 1.16 Gb; scaffold N50 = 32 Mb) (*Richards et al., 2021*) using bwa-mem (v. 0.7.2). Duplicate reads were identified using MarkDuplicates and BAM indices were subsequently generated using the Picard software package (*Broad Institute, 2018*). Samples were genotyped following *Richards et al., 2021*, according to GATK best practices (*DePristo et al., 2011*). Specifically, SNPs were called and filtered using hard-filtering criteria in HaplotypeCaller. We used the following criteria in our filtering approach: QD < 2.0; QUAL < 20; FS < 60; MQRankSum < –12.5; ReadPosRankSum < –8 (*DePristo et al., 2011*; *Poplin et al., 2017*; *Marsden et al., 2014*).

Following initial genotyping with GATK, we subsequently filtered our data further using VCFtools (*Danecek et al., 2011*). Specifically, we filtered using the following flags: --maf 0.05; --min-alleles 2; --max-alleles 2; --min-meanDP 7; --max-meanDP 100; --max-missing 0.85. Indels were removed. To reduce non-independence among sites in our final dataset, we conservatively removed sites in strong linkage disequilibrium using plink v1.9 (--indep-pairwise 10['kb'] 50 0.5: *Purcell et al., 2007*). This resulted in the retention of 1,129,771 SNPs across 139 hybrid samples and the 69 wild-caught samples from *Richards et al., 2021*. Unless otherwise specified, these SNPs were used for all downstream analyses.

### Hybrid fitness measures

We used three proxies for fitness: survival, growth, or a composite measure of the two. Survival was a binary trait indicating whether a fish survived (i.e. a tagged fish was recovered) or not during its exposure period in field enclosures. Growth was a continuous measure, defined as the proportional increase in standard length $\left(\frac{\text{Final SL} - \text{Starting SL}}{\text{Starting SL}}\right)$. Lastly, we defined composite fitness as survival × growth, similar to the metric used in *DiVittorio et al., 2020*, and analogous to composite fitness in

*Hereford, 2009*, who used fecundity as their second fitness measure, rather than growth. Composite fitness is equal to growth for survivors and equals zero for non-survivors because growth could not be assessed for non-surviving individuals. Because composite fitness represents the most information-rich metric of fitness, we report composite fitness results in the main text; results for growth and survival are included in the supplement.

## Population genetic variation

To visualize genetic variation present in hybrids and across lakes (Crescent Pond and Little Lake), we first used a PCA of genetic variation using plink v1.90 (*Purcell et al., 2007*, *Figure 1*), plotting the first two principal component axes using R (version 3.6.3: *R Development Core Team, 2019*). We then estimated admixture proportions in hybrids using ADMIXTURE v1.3.0 (82). Populations of each species were substantially differentiated between Crescent Pond and Little Lake (*Martin and Feinstein, 2014*; *Richards et al., 2021*); thus, independent ADMIXTURE analyses were conducted for each lake. Because we were primarily interested in admixture proportions of hybrids, we set K = 3 in these analyses, corresponding to the three parental species used in hybrid crosses. Using admixture proportions of hybrid individuals, we tested the hypothesis that ancestry predicts hybrid composite fitness in experimental field enclosures by fitting a GAM including either (1) scale-eater ancestry or (2) molluscivore ancestry with fixed effects for experiment and lake. This was repeated for survival and growth separately. Composite fitness was analyzed using a tobit (zero-censored) model to account for zero-inflation using the censReg R package (*Henningsen, 2020*), survival was analyzed using a binomial model, and growth was analyzed using a Gaussian model. We conducted additional ADMIXTURE analyses that either (1) were supervised, with generalist, molluscivore, and scale-eater parentals a priori assigned to one of three populations, with only hybrid ancestry proportions being estimated by admixture, or (2) using only samples from the second field experiment. The same linear models described above were subsequently repeated using these alternative admixture proportions.

## Genome-wide association tests

To identify SNPs that were most strongly associated with fitness (survival, growth, or composite), we implemented the LMM approach in GEMMA (version 0.98.1: *Zhou and Stephens, 2012*). This analysis was repeated using each fitness measure as the response variable. To account for relatedness among samples, we estimated the kinship matrix among all 139 hybrid samples, which in turn were used in downstream LMMs. To account for the potentially confounding effect of year/experiment and lake on estimated fitness measures, we included each as covariates in the LMMs. To ensure rare variants were not included in these analyses, we only included sites that had a minor allele frequency greater than 5% across all hybrids. A total of 933,520 SNPs were analyzed; 196,251 SNPS were excluded due to allele frequency change following removal of parental species. SNPs strongly associated with fitness were identified with (1) an FDR (*Benjamini and Hochberg, 1995*) less than 0.05 or a (2) $p$-value < 0.05 following Bonferroni correction. We focused primarily on the sites identified by the conservative Bonferroni correction, however.

## Gene ontology enrichment

We annotated sites that were significantly associated with fitness using snpEff (*Cingolani et al., 2012*) and the annotated *C. brontotheroides* reference genome (*Richards et al., 2021*). We constructed a custom database within snpEff using the functional annotations originally produced by *Richards et al., 2021*, and subsequently extracted information on the annotations and putative functional consequences of each variant.

Using genes identified for each SNP that was significantly associated with one of the fitness measures, we performed gene ontology enrichment analyses using ShinyGO v0.61 (*Ge et al., 2020*). For genes identified as being intergenic, we included both flanking genes. As in *Richards et al., 2021*, the gene symbol (abbreviation) database that had the greatest overlap with ours was that of the human database; thus, we tested for enrichment of biological process ontologies curated for human gene functions, based on annotations from Ensembl. Results are reported for biological processes that were significantly enriched with FDR < 0.05. We then compared this list of candidate loci to those identified in past studies of San Salvador Island pupfishes (*Richards et al., 2021*; *McGirr and Martin, 2020*; *McGirr and Martin, 2021*).

## Morphometrics

We measured 31 external morphological traits for all 139 hybrids and 69 parental individuals from Crescent Pond (30 generalists, 19 molluscivores, and 30 scale-eaters) and 85 from Little Lake (30 generalists, 25 molluscivores, and 30 scale-eaters). We digitally landmarked dorsal and lateral photographs (both sides) of each lab-reared hybrid (pre-release) or parent using DLTdv8 (*Hedrick, 2008*). Measurements included 27 linear distances and 3 angles. For nearly all individuals, lateral measurements were collected from both lateral photographs and averaged. Morphological variables were size-corrected using the residuals of a $\log_{10}$(trait) ~$\log_{10}$(standard length) regression standardized for selection analyses as outlined in the supplement. We used these 31 morphological traits to estimate two linear discriminant (LD) axes that best distinguished the generalist, molluscivore, and scale-eater using the lda function in the mass package in R. We then used the resultant LD model to predict LD scores for the 139 sequenced hybrids for later use in GAMs.

## Estimation of adaptive landscapes

We fit GAMs using the mgcv package v1.8.28 (*Wood, 2011*) in R to estimate fitness landscapes for the two discriminant axes (LD1–2) and fitness. All models included a thin-plate spline fit to the two LD axes and we included both lake and experiment in all models as fixed effects. Lake by experiment interaction terms were also included in some models. Models were ranked using the corrected Akaike information criterion for small sample sizes (AICc) and were considered to be a substantially worse fit to the data if ΔAICc > 4 from the best-fit model (*Burnham and Anderson, 2002*). The best-fit model from the above approach was in turn used to visualize fitness landscapes, plotting predicted values of fitness measures on the two discriminant axes in R (*Figure 2*).

Using these results, we tested whether inclusion of SNPs that were strongly associated with fitness (i.e. those that surpassed the 0.05 Bonferroni threshold) improved estimation of fitness landscapes. We first extracted genotypes for the highly significant SNPs identified by GEMMA (13 for composite fitness, 4 for only growth: see section *Genome-wide association tests*), and coded these as either reference, single, or double mutants using VCFtools (*Danecek et al., 2011*). We then used the best-fit models identified above and fit a range of models that included one or all SNPs. Individual fitness-associated SNPs were treated as ordered factors (i.e. transition from homozygous reference to heterozygote to homozygous alternate) and modeled using a factor smooth in the GAMs. Note that factor 'smooths' are effectively modeled as step functions.

To quantify whether the local features of the complete fitness landscape constructed using all morphological variables and the most strongly fitness-associated SNPs were robust to sampling uncertainty, we conducted a bootstrapping procedure for this model. Specifically, we resampled hybrids with replacement 10,000 times and refit the full model. We then calculated the mean predicted composite fitness for each LD axis in slices across the fitness landscape, both for our observed dataset and for each bootstrap replicate. Slices divided the fitness landscape into thirds for each LD axis. We then quantified the mean and standard deviation of the predicted composite fitness for each position along the other LD axis.

We quantified uncertainty (mean ± SD) around local features of the bootstrapped fitness landscapes as compared to the observed values of predicted fitness for the same 'slice' of the fitness landscape. We predicted values at the same 30 points along each LD axis. We then plotted the locations of parents along the x-axis (LD1 or LD2) to enable relation of features on the fitness landscape to parental phenotypic distributions.

## Estimation of genotypic fitness networks

We first estimated genotypic networks using sites previously shown to be highly divergent ($F_{ST} > 0.95$) and showing significant evidence of a hard selective sweep in one of the trophic specialists (based on evidence from both SweeD and OmegaPlus: *Richards et al., 2021*; *Pavlidis et al., 2013*; *Alachiotis et al., 2012*). We identified the SNPs in our unpruned full dataset overlapping with sites inferred to have undergone selective sweeps (*Richards et al., 2021*), resulting in 380 SNPs for molluscivores and 1118 SNPs for scale-eaters. We subsequently constructed genotypic fitness networks in igraph v1.2.4.1 (142) following the procedure outlined in the supplement.

To visualize the high-dimensional genotypic fitness network, we randomly sampled 10 adaptive loci 100 times and plotted haplotypes connected by edges if they were within five mutational steps of one

another (*Figure 3C*). Then, we calculated the mean distance between all species pairs (in number of mutational steps). We used pairwise Tukey's HSD tests to test whether inter-species distances differed.

## Estimation of evolutionary accessibility

We tested whether the evolutionary accessibility of genotypic fitness trajectories through observed hybrid genotypes from generalist to each specialist species differed based on the source of genetic variation. We restricted our investigation to networks composed of adaptive loci as previously described (*Figure 3A*: *Richards et al., 2021*). This included a total of 380 SNPs in the molluscivores, and 1118 in the scale-eaters. The reduced number of adaptive SNPs sites in our dataset as compared to that of *Richards et al., 2021* is due primarily to the increased stringency of our filtering. We further partitioned these SNPs by their respective sources: standing genetic variation (molluscivore N = 364; scale-eater N = 1029), de novo mutation (scale-eater N = 24), or introgression (molluscivore N = 16; scale-eater N = 65), again using the assignments from *Richards et al., 2021*. For analyses of trajectories between generalists and molluscivores, we included only SNPs found to be sweeping in molluscivores; likewise, we included only SNPs sweeping in scale-eaters for analysis of trajectories between generalists and scale-eaters.

The full procedure for constructing genotypic fitness networks, identifying accessible paths, and quantifying accessibility is outlined in the supplement. Briefly, we randomly generated 5000 datasets of five SNPs comprised of either (1) standing genetic variation, (2) adaptive introgression, (3) de novo mutation (scale-eaters only), (4) standing genetic variation + adaptive introgression, (5) standing genetic variation + denovo mutation, or (6) standing genetic variation + adaptive introgression + denovo mutation (scale-eaters only). We additionally repeated this procedure using both classes of SNPs for molluscivores to determine whether genotypic trajectories separating generalists to molluscivores are more accessible than those between generalists and scale-eaters. Because different sets of sites are sweeping in each specialist, we conducted these analyses separately for each species. We then constructed genotypic networks, in which nodes are haplotypes of SNPs encoded in 012 format (0 = homozygous reference, 1 = heterozygote, 2 = homozygous alternate), and edges link mutational neighbors. When determining whether a path was accessible or not, we only included paths for which each mutational step (i.e. each intervening haplotype) between generalist to specialist was observed in at least one hybrid sample.

With these networks, we sought to ask (1) whether molluscivores or scale-eaters are more accessible to generalists on their respective genotypic fitness landscapes, (2) whether the ruggedness of the genotypic fitness landscape varied among specialists, and (3) whether accessibility is contingent upon the source of genetic variation available to natural selection. For each random network sampled and for each measure of fitness, we calculated (1) the minimum length of accessible paths between a random generalist and specialist sampled from our sequenced individuals, (2) the number of accessible paths between the same generalist and specialist pair, (3) the number of nodes, (4) the number of edges in the network, (5) the number of peaks on the landscape (genotypes with no fitter neighbors; *Ferretti et al., 2018*), (6) the distance of parental nodes to these peaks, and (7) the number of accessible paths separating them. Larger networks often have a greater number of potential paths, including both accessible and inaccessible paths (*Figure 4—figure supplement 1*), and we were interested in the relative availability of accessible adaptive pathways. Consequently, we divided the number of accessible paths in each random network sampled by the number of nodes. Using our six summary statistics, we tested whether accessibility and landscape ruggedness differed between networks constructed from SGV/introgression/de novo mutations (for scale-eaters) or SGV/introgression (for molluscivores). To do so, we calculated the mean and standard error of each summary statistic across all 5000 replicates. We then modeled the association between each summary statistic and species using a logistic regression, whereby species was modeled as a binary response variable (i.e. scale-eater networks = 0, molluscivore networks = 1), with each measure of accessibility as the predictor. We arbitrarily treated scale-eater networks as the control, and using the estimated coefficients obtained an OR that corresponds to the extent to which molluscivore networks either have increased (OR > 1) or decreased (OR < 1) accessibility measures relative to scale-eater networks. Significance was similarly assessed using a likelihood ratio test. Additional details on this procedure may be found in the supplement. Using the fitted logistic model, we conducted a likelihood ratio test to quantify significance. To explicitly test the hypothesis that increasing landscape ruggedness

reduced the length of accessible paths to the nearest fitness peak, we fit a Poisson regression model in R in which the number of fitness peaks predicts the length of the shortest accessible path between any generalist or specialist node and any fitness peak on that landscape: glm(Min. Distance to Peak ~ Number of Peaks, family = 'poisson').

A similar procedure was used to assess whether measures of accessibility (scaled number of accessible paths, length of the shortest accessible path) and landscape ruggedness (number of peaks) differed within species among networks constructed from different sources of genetic variation. Here, networks constructed from SGV were treated as the control, to which all other networks were compared. For example, to test whether accessibility of the generalist-to-scale-eater paths are greater in networks constructed from de novo mutations than those from SGV, a logistic model was fitted wherein the response variable for SGV networks was assigned to be 0, and 1 for de novo networks. As before, significance was similarly assessed using a likelihood ratio test, but here *p*-values were corrected for multiple testing using the FDR (*Benjamini and Hochberg, 1995*). We assessed whether differences in these measures among the two alternate generalist to specialist trajectories in networks constructed from all three sources of variation were significant using an ANOVA in R (*R Development Core Team, 2019*). Due to the highly skewed nature of these distributions, post hoc pairwise significance was assessed using a nonparametric Kruskal-Wallis one-way analysis of variance in the agricolae package (*de Mendiburu, 2020*) in R.

## Acknowledgements

We thank Michelle St. John, David Tian, Jacqueline Galvez, and Maria Fernanda Palominos for insightful discussion of the results; the Gerace Research Centre and Troy Day for logistical support; the BEST Commission and the government of the Bahamas for permission to collect, import, and export samples; the Vincent J Coates Genomics Sequencing Center and Functional Genomics Laboratory at UC Berkeley for performing whole-genome library preparation and sequencing (supported by NIH S10 OD018174 Instrumentation Grant), and the University of California, Berkeley, and University of North Carolina at Chapel Hill for computational resources. This work was funded by the National Science Foundation DEB CAREER grant #1749764, National Institutes of Health grant 5R01DE027052-02, the University of North Carolina at Chapel Hill, and the University of California, Berkeley to CHM, graduate research funding from the Museum of Vertebrate Zoology to EJR, and an NSF Postdoctoral Research Fellowship in Biology under Grant No. 2109838 to AHP.

## Additional information

### Funding

| Funder | Grant reference number | Author |
| --- | --- | --- |
| National Institutes of Health | 5R01DE027052-02 | Christopher H Martin |
| National Science Foundation | 1749764 | Christopher H Martin |
| National Science Foundation | 2109838 | Austin H Patton |
| University of North Carolina at Chapel Hill | | Christopher H Martin |
| University of California, Berkeley | | Christopher H Martin |

The funders had no role in study design, data collection and interpretation, or the decision to submit the work for publication.

### Author contributions

Austin H Patton, Conceptualization, Data curation, Formal analysis, Funding acquisition, Investigation, Methodology, Resources, Software, Visualization, Writing – original draft, Writing – review

and editing; Emilie J Richards, Formal analysis, Writing – original draft, Writing – review and editing; Katelyn J Gould, Logan K Buie, Data curation, Writing – original draft, Writing – review and editing; Christopher H Martin, Conceptualization, Data curation, Funding acquisition, Investigation, Methodology, Project administration, Resources, Supervision, Validation, Writing – original draft, Writing – review and editing

**Author ORCIDs**
Austin H Patton ⓘ http://orcid.org/0000-0003-1286-9005
Emilie J Richards ⓘ http://orcid.org/0000-0003-2734-6020
Katelyn J Gould ⓘ http://orcid.org/0000-0001-5207-1810
Christopher H Martin ⓘ http://orcid.org/0000-0001-7989-9124

**Ethics**
All experiments were carried out following approved protocols from the University of California, Davis Institutional Animal Care and Use Committee (#17455), the University of North Carolina at Chapel Hill Animal Care and Use Committee (#18-061.0), and the University of California, Berkeley Animal Care and Use Committee (AUP-2015-01-7053).

**Decision letter and Author response**
Decision letter https://doi.org/10.7554/eLife.72905.sa1
Author response https://doi.org/10.7554/eLife.72905.sa2

## Additional files

**Supplementary files**
• Supplementary file 1. Supplementary tables. (a)—Table 1. Samples of hybrids and parental studies used either in genomic or in morphological analyses, along with associated metadata. (b)—Table 2. Models tested to assess the extent to which specialist ancestry predicts measures of fitness and their respective fits using all samples and an unsupervised ADMIXTURE analysis. Best-fit models are bolded. (c)—Table 3. Models tested to assess the extent to which specialist ancestry predicts measures of fitness and their respective fits using all samples and a supervised ADMIXTURE analysis. Best-fit models are bolded. (d)—Table 4. Models tested to assess the extent to which specialist ancestry predicts measures of fitness and their respective fits using only samples from the second field experiment (*Martin and Gould, 2020*) and an unsupervised ADMIXTURE analysis. Best-fit models are bolded. (e)—Table 5. Models tested to assess the extent to which genome-wide variation (PC1/PC2) predicts measures of fitness and their respective fits using all samples and an unsupervised ADMIXTURE analysis. Best-fit models are bolded. (f)—Table 6. Single nucleotide polymorphisms (SNPs) found to be strongly associated with composite fitness using SnpEff (*Cingolani et al., 2012*). SNPs that were identified as being strongly associated with both growth and composite fitness are italicized, and those that remain significant after a Bonferroni correction are bolded. (g)—Table 7. Gene ontology term enrichment for genes associated with composite fitness. (h)—Table 8. SNPs found to be strongly associated with growth SnpEff (*Cingolani et al., 2012*). SNPs that were identified as being strongly associated with both growth and composite fitness are italicized, and those that remain significant after a Bonferroni correction are bolded, (i)—Table 9. Gene ontology term enrichment for genes associated with growth. (j)—Table 10. List of the 31 morphological traits measured for this study and standard length; corresponding landmark IDs match those shown in *Figure 2—figure supplement 3*. (k)—Table 11. Generalized additive models fitted to composite fitness. Model fit was assessed using AICc, and Akaike weights represent proportional model support. A thin-plate spline for the two linear discriminant axes s(LD1, LD2) is always included, as is a fixed effect of either experiment (i.e. *Martin and Wainwright, 2013a*; *Martin and Gould, 2020*) or lake (Crescent Pond/Little Lake) or an interaction between the two. In the last two models, experiment and lake are included as splines, modeled using a factor smooth (bs = 'fs'). The best-fit model had five estimated degrees of freedom. (l)—Table 12. Generalized additive models fitted to growth. Model fit was assessed using AICc, and Akaike weights represent proportional model support. A thin-plate spline for the two linear discriminant axes s(LD1, LD2) is always included, as is a fixed effect of either experiment (i.e. *Martin and Wainwright, 2013a*; *Martin and Gould, 2020*) or lake (Crescent Pond/Little Lake) or an interaction between the two. In the last two models, experiment and lake are included as splines, modeled using a factor smooth (bs = 'fs'). The best-fit model had 8.93 estimated degrees of freedom. (m)—Table 13. Generalized additive models fitted to survival. Model fit was assessed

using AICc, and Akaike weights represent proportional model support. A thin-plate spline for the two linear discriminant axes s(LD1, LD2) is always included, as is a fixed effect of either experiment (i.e. *Martin and Wainwright, 2013a*; *Martin and Gould, 2020*) or lake (Crescent Pond/Little Lake) or an interaction between the two. In the last two models, experiment and lake are included as splines, modeled using a factor smooth (bs = 'fs'). The best-fit model had five estimated degrees of freedom. (n)—Table 14. Generalized additive models fitted to growth including SNPs most strongly associated with composite fitness. Model fit was assessed using AICc, and Akaike weights represent proportional model support. The best-fit model for composite fitness using morphology alone (see Table 8) was used as the base model. The SNPs that were most strongly associated with composite fitness (following a Bonferroni correction) were included as fixed effects, modeled as splines using a factor smooth, treating genotype as an ordered factor. Note that three SNPs were excluded due to their close proximity to other SNPs that were more strongly associated. All SNPs were considered individually, as well as all SNPs together. We were unable to assess all possible combinations of SNPs due to the vast number of potential models given the number of SNPs under consideration; rather, we fit one final model that only included SNPs found to be significant in the full model. In turn this model led to a substantial improvement in AICc. The best-fit model had 20.29 estimated degrees of freedom. (o)—Table 15. Generalized additive models fitted to growth including SNPs most strongly associated with growth. Model fit was assessed using AICc, and Akaike weights represent proportional model support. The best-fit model for growth using morphology alone (see Table 9) was used as the base model. Each of the four SNPs that were most strongly associated with growth (following a Bonferroni correction) were included as fixed effects, modeled as splines using a factor smooth, treating genotype as an ordered factor. All SNPs were considered individually, as well as all possible combinations. This was only feasible due to the small number of SNPs assessed (four). The best-fit model had 7.97 estimated degrees of freedom. (p)— Table 16. General linear models fitted to examine the relationship between aspects of network size (i.e. number of nodes, number of edges linking neighboring nodes) and the number of accessible paths between generalists and specialists. Models were fitted using each of the three different fitness measures; bolded lines correspond to the best-fit model for each response variable, within each measure of fitness. Poisson regression was chosen as each response variable correspond to count-data. Because Poisson regression models are log-linear, we report both the estimated coefficient and its exponentiated value which corresponds to the expected multiplicative increase in the mean of Y per unit value of X. (q)—Table 17. Accessibility of specialists to generalists and the ruggedness of their respective fitness landscapes. Odds ratios were obtained by modeling the association between each summary statistic and the species from which adaptive loci were used to construct the fitness network. Scale-eaters were treated as the baseline of comparison in the comparison of odds ratios; thus, positive odds ratios imply that summary statistics for molluscivore fitness networks are greater than those constructed from scale-eater adaptive loci and vice versa. For generalist to specialist comparisons, accessible paths were identified between one randomly sampled generalist node and one randomly sampled specialist node. For comparison of the peaks in networks, these summary statistics were calculated from either molluscivore or scale-eater fitness networks, identifying the number of peaks (nodes with no fitter neighbors – see Materials and methods), and the scaled (total divided by number of nodes in the network) number of accessible paths separating all focal specialist nodes and all peaks in the network. (r)—Table 18. Influence of different sources of adaptive genetic variation on accessibility of fitness paths separating either generalists from molluscivores, or generalists and scale-eaters using all samples. Results for networks using all three measures of fitness (composite fitness, survival, and growth) are reported. Networks were constructed from random draws of five SNPs from either standing genetic variation (SGV), introgression, or de novo mutations, as well as their combinations. Odds ratios were obtained by modeling the association between each accessibility measure and the source of genetic variation used to construct the fitness network, relative to networks constructed from standing variation. Thus, positive odds ratios imply that networks from standing variation have measures of accessibility that are smaller as compared to the alternative (e.g. introgression, de novo mutations, etc.). (s)—Table 19. Influence of different sources of adaptive genetic variation on accessibility of fitness paths separating either generalists from molluscivores, or generalists and scale-eaters using only samples from the second field experiment (*Martin and Gould, 2020*). Results for networks using all three measures of fitness (composite fitness, survival, and growth) are reported. Networks were constructed from random draws of five SNPs from either standing genetic variation (SGV), introgression, or de novo mutations, as well as their combinations. Odds ratios were obtained by modeling the association between each accessibility measure and the source of genetic variation used to construct the fitness network, relative to networks constructed from

standing variation. Thus, positive odds ratios imply that networks from standing variation have measures of accessibility that are smaller as compared to the alternative (e.g. introgression, de novo mutations, etc.).

- Transparent reporting form

## Data availability

Genomic data are archived at the National Center for Biotechnology Information BioProject Database (Accessions: PRJNA690558; PRJNA394148, PRJNA391309, PRJNA841640). Sample metadata including morphological measurements and admixture proportions have been uploaded to dryad: https://doi.org/10.5061/dryad.0vt4b8h0m. Associated scripts used to estimate genotypic and phenotypic fitness landscapes are hosted at the following github repository: https://github.com/austinhpatton/Pupfish-Fitness-Landscapes (copy archived at swh:1:rev:27e7640ba769886af9ef0b2e6d6f522c9f26e2df).

The following datasets were generated:

| Author(s) | Year | Dataset title | Dataset URL | Database and Identifier |
|---|---|---|---|---|
| Patton AH, Richards E, Gould KJ, Buie LK, Martin CH | 2021 | Hybridization alters the shape of the genotypic fitness landscape, increasing access to novel fitness peaks during adaptive radiation | https://dx.doi.org/10.5061/dryad.0vt4b8h0m | Dryad Digital Repository, 10.5061/dryad.0vt4b8h0m |
| Patton AH, Richards E, Gould KJ, Buie LK, Martin CH | 2022 | Hybridization alters the shape of the genotypic fitness landscape, increasing access to novel fitness peaks during adaptive radiation | https://www.ncbi.nlm.nih.gov/bioproject/PRJNA841640 | NCBI BioProject, PRJNA841640 |

The following previously published datasets were used:

| Author(s) | Year | Dataset title | Dataset URL | Database and Identifier |
|---|---|---|---|---|
| University of North Carolina at Chapel Hill | 2017 | Craniofacial divergence in Caribbean Pupfishes | https://www.ncbi.nlm.nih.gov/bioproject/?term=PRJNA391309 | NCBI BioProject, PRJNA391309 |
| University of North Carolina at Chapel Hill | 2017 | Adaptive introgression contributes to microendemic radiation of Caribbean pupfishes | https://www.ncbi.nlm.nih.gov/bioproject/?term=PRJNA394148 | NCBI BioProject, PRJNA394148 |
| University of California, Berkeley | 2021 | Adaptive radiation of Caribbean pupfish is assembled from an ancient and disparate spatiotemporal landscape | https://www.ncbi.nlm.nih.gov/bioproject/?term=PRJNA690558 | NCBI BioProject, PRJNA690558 |

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

## Appendix 1

### Supplementary methods

#### Sampling of hybrid individuals

Samples of hybrid *Cyprinodon* pupfish included herein were first collected following two separate fitness experiments, conducted on San Salvador Island in 2011 (*Martin and Wainwright, 2013a*) and 2016 (*Martin and Gould, 2020*), respectively. Experiments were carried out in two lakes: Little Lake (LL), and Crescent Pond (CP). Following their initial collection at the conclusion of their respective experiments (see *Martin and Wainwright, 2013a*, and *Martin and Gould, 2020*, for protocols), samples were stored in ethanol. In late 2018, 149 hybrid samples were selected for use in this experiment. Of these, 27 are from the experiment conducted in 2011 (14 from LL, 13 from CP), and the remaining 122 are from the 2016 experiment (58 from LL, 64 from CP). Due to reduced sample size for some species within Little Lake, we include fish obtained from Osprey Lake for downstream analyses comparing hybrids to Little Lake, as the two comprise a single, interconnected body of water.

#### Genomic library prep

DNA was extracted from the muscle tissue of hybrids using DNeasy Blood and Tissue kits (Qiagen, Inc); these extractions were then quantified using a Qubit 3.0 fluorometer (Thermo Scientific, Inc). Genomic libraries were prepared by the Vincent J Coates Genomic Sequencing Center (QB3) on the automated Apollo 324 system (WaterGen Biosystems, Inc). Samples were fragmented using a Covaris sonicator, and barcoded with Illumina indices. Samples were quality checked using a Fragment Analyzer (Advanced Analytical Technologies, Inc). All samples were sequenced to approximately 10× raw coverage on an Illumina NovaSeq.

#### Genotyping and filtering

Because we are interested in genomic variants found not only within our hybrid samples but across San Salvador Island and the Caribbean, we conducted genotyping including all 247 samples from *Richards et al., 2021*. These samples include members of the three species (*C. variegatus*, *C. brontotheroides*, *C. desquamator*) found on San Salvador Island, as well as individuals of *C. variegatus* found throughout the Caribbean, and numerous outgroups (*C. laciniatus*, *C. higuey*, *C. dearborni*, *Megupsilon aporus*, and *Cualac tesselatus*). We then excluded *M. aporus* and *C. tesselatus* along with 18 additional samples for which necessary data for downstream analyses were missing (e.g. quality photographs for the collection of morphological data). This approach led to the retention of a total of 4,206,786 total SNPs and 139 hybrid individuals.

#### Morphometrics

Because the morphological measurements used in the 2013 and 2016 experiments differ slightly, we remeasured all sequenced hybrid individuals and up to 30 individuals of each parent species for the 30 morphological characters described in *Martin and Gould, 2020*, as well as for standard length (SL).

For each photograph, unit-scale was obtained by additionally landmarking points on a regular grid included in each photograph using DLTdv8a (*Hedrick, 2008*). Landmark data (x-y coordinates in units of pixels) were subsequently uploaded into R and converted to millimeters (in the case of linear measurements) or degrees (for angular measurements) using a custom script.

We then assessed, for each trait, the need for size correction. That is, we sought to avoid an outsized role of body size in downstream interpretation. Thus, if a trait was colinear with SL, we regressed the two (treating SL as the predictor) and took the residuals. In each case, both SL and the response were log10 transformed. Subsequently, residuals were scaled such that their distribution had a mean of 0, and a standard deviation of 1. Traits that did not need size correction were also log-transformed and unit-scaled.

We used these morphological measurements to estimate two LD axes that distinguish the generalist, molluscivore, and scale-eater using the LDA function in R. That is, we used morphological data from the 165 parental fish to estimate LD scores for each individual. Doing so, we were able to correctly assign individual fish to their corresponding species with 99.4% accuracy (*Figure 2— figure supplements 4–5*). Attempting to predict species assignment by lake did not improve this

prediction accuracy (instead reducing prediction accuracy to 98.7%); consequently, we proceeded with the LD axes estimated without accounting for lake.

We additionally asked whether (1) specialists were more morphologically constrained than generalists and (2) if hybrids were less constrained than the three parental species. To do so, we calculated morphological disparity per group using the dispRity.per.group function implemented in the R package dispRity v1.3.3 (*Guillerme and Poisot, 2018*). Specifically, this function calculates the median distance between each row (sample) and the centroid of the matrix per group. By bootstrapping the data 100 times, we tested the hypothesis that each of the three parental species differed significantly in their morphological disparity, first using an ANOVA, followed by a post hoc pairwise t-test to assess pairwise significance in R. We corrected for multiple tests using an FDR correction.

To test whether genetic distance predicts morphological distance, we calculated two distance matrices. First, we calculated pairwise Euclidean distances between all hybrids using all morphological variables. Then, we calculated pairwise genetic distances between all hybrids using the final set of SNPs described above with the genet.dist function implemented in vcfR (*Knaus and Grünwald, 2017*). We then fit a simple linear model, regressing genetic distance on morphological distance.

## Estimation of adaptive landscapes

We sought to characterize the extent to which the three measures of fitness are predicted by morphology alone and to, in turn, visualize fitness landscapes for our sequenced hybrids. To do so, we fitted six GAMs using the mgcv package v1.8.28 (*Wood, 2011*) in R. All models included a thin-plate spline fitted for the two LD axes. Because we have strong a priori knowledge that fitness outcomes will be contingent to some extent on experiment year and on the lake in which hybrids were placed, we include experiment and lake in all fitted models, modeled either as fixed effects or as an interaction term between the two. Additionally, we fitted models that included individual splines for each LD axis, either with or without experiment or lake as a factor smooth. The full list of models and their respective fits are included in the supplement (*Supplementary file 1–tables 11–13*).

For composite fitness, we excluded three SNPs that were within close proximity (i.e. <1000 bp) to an SNP that was more significantly associated. Because of the reduced number of significantly associated SNPs identified for growth (4) as compared to composite fitness (10), we were able to fit and compare all combinations of significantly associated SNPs for the former. The best-fit model for composite fitness was also the most complex, including all fitness-associated SNPs. Thus, to reduce model complexity, we fit one additional model, excluding any of the three SNP (fixed effect) that was not significant in the full model. The full range of models and their associated fits are reported in *Supplementary file 1–tables 14–15*. As before, predicted fitness values across LD space were extracted from the best-fit model and plotted using R.

## Genotypic fitness networks

Recent work has shown that the adaptive radiation of the pupfish of San Salvador Island involved selection on standing genetic variation, adaptive introgression, and de novo mutation (*Richards et al., 2021*). Furthermore, the specialists on San Salvador Island received approximately twice as much adaptive introgression as did generalists on neighboring islands. These findings imply that each source of genetic variation may exert a unique influence on the fitness landscape, in turn facilitating the radiation. We sought to explore this possibility and so estimated genotypic fitness networks using sites previously shown to have undergone hard selective sweeps in specialists. To do so, we identified the SNPs in our un-thinned dataset overlapping with sites inferred to have undergone selective sweeps (*Richards et al., 2021*) to produce two datasets (one for each specialist). We then constructed networks using the following procedure:

1. Target SNPs were extracted from the vcf file of all sequenced hybrids and parental species from Crescent Pond and Little Lake in 0/1/2 format using VCFtools. That is, individuals genotyped as homozygote reference were coded as 0, heterozygotes as 1, and homozygote alternatives as 2. Note again that the reference genome used was that of the molluscivore, *C. brontotheroides*.
2. These SNPs were subsequently loaded into R, and concatenated into haplotypes, such that each sequenced individual has a haplotype. These per-sample haplotypes were subsequently

associated with metadata, namely the observed binary survival, growth, assignment as hybrid or one of the three parental species, and lake of origin.

3. Each haplotype was subsequently collapsed and summarized, such that mean survival, growth, and composite fitness are retained for each unique haplotype. Additionally, the number of hybrids, generalists, molluscivores, and scale-eaters that have the haplotype were recorded.

4. The distance in number of mutational steps was then calculated for each pairwise combination of haplotypes. For example, the distance between haplotype 000 and 001 is a single mutational step, whereas haplotypes 000 and 002 are two steps away.

5. Lastly, we constructed networks using the R package igraph v1.2.4.1 (*Csardi and Nepusz, 2006*). Specifically, nodes represent haplotypes, and edges are drawn between haplotypes that are mutational neighbors (i.e. are a single mutational step away). Nodes present in hybrids were colored and sized proportional to their respective mean fitness. Nodes present only in parental species were colored according to the species in which that haplotype is unique to.

## Estimation of evolutionary accessibility

The large number of SNPs in our dataset above raises numerous challenges in the visualization and summarization of fitness networks. Perhaps most significant is that, as the number of sites assessed increases, the sequence space increases vastly; the number of potential haplotypes is defined by 3 to the power of the number of SNPs, and the number of potential edges is defined by the number of haplotypes choose 2. Consequently, networks constructed from a larger number of focal SNPs are comprised of haplotypes that are separated on average by more mutational steps than those constructed from fewer SNPs. Because we can only interpret the fitness consequences of evolutionary trajectories for which we have data along each mutational step, we restricted analysis to haplotypes that are mutational neighbors (i.e. separated by a single mutational step).

To do so, we developed a permutation approach to construct fitness networks from SNPs that were sourced from the three sources of genetic variation defined above as well as all possible combinations including standing variation (i.e. standing genetic variation + adaptive introgression and/or de novo mutation). Specifically, from each set we sampled five SNPs up to 5000 times. For networks constructed from combinations of sources (e.g. SGV + introgression), we ensured that at least one of each source was present. To do so, we generated 1000 random sets of SNPs for all possible combinations (e.g. 1 SGV – 4 introgression, 2 SGV – 3 introgression, etc.). Then, we sampled up to 5000 of these combinations; these samples comprise our permutations. Then, from each permutation, we constructed fitness networks using the five steps defined above; these networks served as the subsequent assessment of evolutionary accessibility of genotypic trajectories separating the generalists from either specialist. Edges were only drawn such that the network is directed; that is, edges were drawn from low to higher or equal fitness nodes.

We constructed networks using both parental species and hybrids. For each trajectory under consideration (generalist → molluscivore and generalist → scale-eater), we first identified all generalist to specialist trajectories that are connected by accessible (monotonically increasing) paths. From these connected generalist-specialist node pairs, we randomly sampled a single pair. We then identified the number and length of accessible paths separating the generalist node from the specialist node and recorded these values using the 'all_simple_paths' function in igraph and excluded any paths that traversed through haplotypes not found in any hybrid (i.e. exclusive to parental species) and thus had no information on fitness.

Specifically, two node paths (a single mutational step from generalist to specialist) were allowed only if both parental nodes (SNP haplotypes) were also observed in hybrids, with fitness data. Three-node paths were allowed if hybrid fitness data was present for at least one of the parental nodes and the intervening node between the two parental nodes. Paths that were four nodes or longer were allowed only if all intervening nodes between parental nodes had associated hybrid fitness data.

We additionally calculated the number of peaks on the genotypic fitness landscape, as well as the number of accessible paths between parental nodes and these peaks, and the minimum distance from parental nodes to a peak on the landscape. We define 'peaks' on the genotypic fitness landscape as genotypes (SNP haplotypes) with no-fitter neighboring genotype follow the definition of *Ferretti et al., 2018*. This definition is inclusive of nodes/genotypes that are equal in fitness to their neighbors, which may be fitter than all other neighboring nodes. We conservatively excluded nodes that shared only a single neighbor.

## Supplementary results

### Population ancestry associations with fitness

When repeating our test for an association between fitness measures and ancestry proportions as estimated from a supervised ADMIXTURE analyses, we recovered similar results with one exception; generalist ancestry was significantly associated with growth rate (generalist: $p = 0.021$). Admixture proportions estimated from an unsupervised analysis did not significantly predict any measure of fitness when only using hybrids from the second field experiment (*Supplementary file 1–table 4*; *Martin and Gould, 2020*). Genome-wide PC1 was associated with composite fitness ($p = 0.004$) and survival ($p < 0.001$), whereas PC2 was not (*Supplementary file 1–table 5*). However, PC1 largely explains differences among lakes (*Figure 1c*); thus, the positive correlation between PC1 and fitness is likely explained by the previously described overall differences in survival observed between the two lakes in past experiments (*Martin and Wainwright, 2013a*; *Martin and Gould, 2020*).

### Genomic associations recovered for composite fitness and growth, not survival

Whereas we identified 132 SNPs that were associated with composite fitness, only 58 were associated with growth and none were associated with survival. Of the SNPs associated with growth, only four remained significant using the conservative Bonferroni threshold. Across all significant sites (either via FDR or Bonferroni correction), a total of 11 were shared across analyses. The only gene proximate to a growth-associated SNP was *csad*. Lastly, we found a single gene shared between our study and the 125 ecological DMIs (putative genetic incompatibilities that are differentially expressed among specialists and misexpressed in F1 hybrids) presented in *McGirr and Martin, 2020*. This gene, associated with growth (but not composite fitness) in our study, is *mettl21e*.

When considering SNPs found to be associated with growth, we did not identify any gene ontologies that were significantly enriched at an FDR < 0.01. However, looking at those enriched at an FDR < 0.05, we do observe a number of ontologies related to biosynthetic processes, and regulation of metabolic processes (*Figure 2—figure supplement 3*). Specifically, the greatest (and most significant) enrichment was for that of phosphorus and phosphate-containing compound metabolic processes and their regulation. Phosphorous deficiencies have previously been associated with poor growth in silver perch (*Bidyanus bidyanus*: *Yang et al., 2006*) and skeletal deformaties (including vertebral compression and craniofacial deformaties) in zebrafish (*Danio rerio*: *Costa et al., 2018*). Similarly, blunt snout bream (*Megalobrama amblycephala*) exhibited greater growth rates with increasing phosphorous levels in their diets (*Yu et al., 2020*). In short, enrichment of growth-associated SNPs for ontologies pertaining to phosphorous metabolism is consistent with the substantial literature documenting that phosphorous availability and metabolism is a determinant of growth in fishes.

### Morphological variation within sampled hybrids

As in the previous two experiments, there is a relative paucity of hybrids exhibiting the morphologies that characterize either specialist. Rather, most hybrids fall near the generalists, with a number exhibiting transgressive morphologies (*Figure 2—figure supplement 5a*). As expected, both specialists exhibit reduced morphological disparity as compared to generalists, and hybrids show the greatest (*Figure 2—figure supplement 5b*). That is, the specialists appear more morphologically constrained than generalists, falling on average closer to the group centroid. Interestingly molluscivores exhibit the least disparity, even less so than scale-eaters.

### Fitness-associated SNPs influence shape of the adaptive landscape

Using morphology alone, the best-fit GAM for survival, growth, and composite fitness were simpler than the model for composite fitness using both morphology and fitness-associated SNPs (*Supplementary file 1–tables 11–13*). For survival and composite fitness (*Figure 2—figure supplement 6a–b*), this model included a thin-plate spline for LD1 and LD2, with experiment and lake included as fixed effects. The resultant landscape was also similar for these two analyses, supporting an interpretation of directional selection, favoring molluscivores. For growth, the best-fit model had the thin-plate spline for LD1 and LD2, but included an interaction term between experiment and lake (*Figure 2—figure supplement 6c*; *Supplementary file 1–table 12*). In contrast to the previous two models, the landscape predicted using growth as our proxy of fitness supported an interpretation

of directional selection in favor or hybrids most similar to generalists, and to a lesser extent, scale-eaters.

Notably, model selection using AICc invariably supported the inclusion of fitness-associated SNPs for growth and composite fitness (*Supplementary file 1–tables 14–15*). For growth, the best-fit model including genotypes was an improvement of 22.99 AICc over the model including morphology alone, whereas for composite fitness, the improvement was 94.527 AICc. Interestingly, the best-fit models and growth including associated SNPs was similar to that of the landscape without fitness-associated SNPs, but largely supported an interpretation of directional selection in favor of scale-eaters, and to a lesser extent generalists.

