## [Editor Report]

This study reports on the inference of the evolutionary trajectory of two specialist species that evolved from one generalist species. The process of speciation is explained as an adaptive process and the changing genetic architecture of the process is analyzed in great detail. The genomic dataset is big and the inference from it is solid. The authors reach the conclusion that introgression and de novo mutations played a major role in this adaptive process.

---

## [Decision Letter]

**Decision letter after peer review:**

Thank you for submitting your article "The first genotypic fitness network in a vertebrate reveals that hybridization increases access to novel fitness peaks" for consideration by *eLife*. Your article has been reviewed by 3 peer reviewers, one of whom is a member of our Board of Reviewing Editors, and the evaluation has been overseen by Molly Przeworski as the Senior Editor. The reviewers have opted to remain anonymous.

Essential revisions:

1) The entire paper is very specific to the system investigated and little attempts are made to reach out to other systems. The articles would gain strongly in doing so. I part the problem may be that the Results and Discussion is merged. This does not invite for a wider discussion. Separating the Results section from the discussion would work well here, or, alternatively one could suggest to have a section called "Results and specific discussion" and a section "General discussion". Clearly the later is largely missing. The conclusion does not rescue this.

2) Data analysis: at several places the reviewers aired concerns about the analysis, both regarding the analysis itself as well as the explanation and justification of it (e.g. Admixture analysis, use of composite fitness, pooling of low and high density treatments). Details are found in the individual reviews below.

*Reviewer #1 (Recommendations for the authors):*

It is of little interest to learn that this is the first study doing exactly this type of work. In one way or the other, every study is unique, no need to stress this in several places, starting from the title(!), abstract, significance,.. conclusion.

The section about calculating the "composite fitness" (= survival * growth) seems odd to me. Survival is 0 or 1, and growth is a quantitative trait. If I understand correctly, growth cannot be assessed when the fish did not survive, so fitness would be NA (because growth is NA). Better describe it: composite fitness is equal to growth, unless the fish died, then it is zero.

How was "composite fitness" statistically analyzed. Tables S2, S8, S11 not allow me to reconstruct this. The distribution of the composite fitness data must be strongly zero inflated, making is hard to analyze them with linear models. I assume that the growth data were more or less normally distributed and survival data binary (0/1), right?

The assumption that mutations are de novo, when they are not known to occur otherwise in samples investigated from San Salvador Island, should be placed by a more realistic framework. There are certainly many rare alleles that are part of the standing genetic variation, but have previously not been observed. The first time they are seen, may suggest that they are novel mutations.

Expressing differences in percent is tricky and potentially open to missunderstandings (e.g. 126 % more accessible). I suggest to use expressions that are more clear (e.g. something like an odd ratio).

Figure 4a. what is the meaning of the different sizes of the coloured dots.

*Reviewer #2 (Recommendations for the authors):*

L195-196: I wish the authors could expand a bit on this statement.

L 412: "the frequency of rare transgressive hybrid phenotypes was altered" I am not sure what the authors are trying to convey here.

Figure 2: l180, it should be panel d) and not c).

Citation 66 is a duplicate of 18.

19 papers co-authored by the last author were cited in the current manuscript. While they all seemed relevant, it still is a lot.

*Reviewer #3 (Recommendations for the authors):*

Line 32: I think whether fitness valleys are crossed is also controversial.

Line 42: "most isolated" Compared to what? Or is this supposed to say mostly isolated?

Lines 73-74: I think usually "accessible" is used to refer to peaks not trajectories, though I can't say your usage is incorrect.

Line 77: The link between monotonically increasing fitness and plausibility of evolving is much broader than Fisher's geometric model; I think it would be better to just state the simple idea that, in large populations, paths with increasing fitness are taken much more readily than alternatives with neutral or deleterious steps.

Line 102: Does this mean stable across all these axes of variation?

Lines 91-110: I'm not clear on how the fitness landscapes in previous work will be distinct from this work, and what parts of Figure 1 support these statements; I suggest a little expansion and reorganization to clarify these issues.

Why is Figure 3 cited in the paper before Figure 2?

I would suggest explicitly defining the fitness measurement in the Results and Discussion, both to make the paper more readable in the current section order and to acknowledge the limitation that reproductive success isn't a part of the composite fitness measurement.

---

## [Author Response]

Essential revisions:1) The entire paper is very specific to the system investigated and little attempts are made to reach out to other systems. The articles would gain strongly in doing so. I part the problem may be that the Results and Discussion is merged. This does not invite for a wider discussion. Separating the Results section from the discussion would work well here, or, alternatively one could suggest to have a section called "Results and specific discussion" and a section "General discussion". Clearly the later is largely missing. The conclusion does not rescue this.

We thank the editor for this suggestion and now have substantially expanded our introduction and Discussion sections. We split the Results and Discussion section as suggested. We generalized the introduction and discussion to address a broader audience and extended our discussion of other systems, as well as the relation of our work to fitness landscape theory

2) Data analysis: at several places the reviewers aired concerns about the analysis, both regarding the analysis itself as well as the explanation and justification of it (e.g. Admixture analysis, use of composite fitness, pooling of low and high density treatments). Details are found in the individual reviews below.

We address these suggestions below. We include new admixture analyses, additional analyses and discussion of our composite fitness measure (as well as the two fitness measures of survival and growth and comparison of the similarities across all three), and new analyses of high and low density fitness experiments analyzed separately to address any concerns about pooling. Our conclusions remained qualitatively unchanged and in some cases our results were strengthened by these additional analyses, detailed below.

We conducted expanded analyses in response to reviewer two’s request for additional investigation of the shape of the fitness landscapes/fitness peaks. Specifically, we now quantify the number of fitness peaks, a classic measure of epistasis and fitness landscape ruggedness (Ferreti et al., 2016, Journal of Theoretical Biology). Consistent with our finding that adaptive introgression and de novo mutations increased accessibility of interspecific genotypic trajectories as compared to standing variation, we also found that landscape ruggedness was reduced. We now discuss the importance of these findings in the context of fitness landscape theory, epistasis, and their relevance the process of speciation.

Reviewer #1 (Recommendations for the authors):It is of little interest to learn that this is the first study doing exactly this type of work. In one way or the other, every study is unique, no need to stress this in several places, starting from the title(!), abstract, significance,.. conclusion.

We have tempered our emphasis of this point, removing it from the title, and completely removing or limiting our discussion of this point throughout in the manuscript.

The section about calculating the "composite fitness" (= survival * growth) seems odd to me. Survival is 0 or 1, and growth is a quantitative trait. If I understand correctly, growth cannot be assessed when the fish did not survive, so fitness would be NA (because growth is NA). Better describe it: composite fitness is equal to growth, unless the fish died, then it is zero.

This interpretation is correct – we have revised our language as recommended to make the interpretation of composite fitness clearer to the reader: Lines 179-181, 560-562.

How was "composite fitness" statistically analyzed. Tables S2, S8, S11 not allow me to reconstruct this. The distribution of the composite fitness data must be strongly zero inflated, making is hard to analyze them with linear models. I assume that the growth data were more or less normally distributed and survival data binary (0/1), right?

This is correct. We now more adequately analyze these data to account for zero inflation (Lines 577-580). Specifically, we now model survival using a binomial model and composite fitness using a tobit model, which accounts for the zero-inflation in our composite fitness measure. Growth is modeled using a gaussian model as before when analyzing the relationship between composite fitness and admixture proportions (Supplementary file 1). These tables are also more explicit with respect to model fit and the statistical distribution used to model each fitness measure. We also report the coefficient to aid in interpretation.

Note that these analyses are further expanded following suggestions from reviewer 2. We now conducted the following expanded analyses (described lines 174 – 177, 580 – 584):

1) an unsupervised admixture analysis (Supplementary file 1),

2) all samples using a supervised admixture analysis (i.e. model is informed a priori which samples are known to belong to either of the three assumed populations/parental species: Supplementary file 1),

3) only samples from the second field experiment (Martin and Gould 2020) for which lake was not found to significantly affect fitness using an unsupervised analysis (Supplementary file 1).

Note that even with the expanded analyses, results are qualitatively the same; ancestry proportions do not strongly influence fitness in this system (lines 190-193, 1600-1605). There is one exception – generalist ancestry appears to positively predict growth when modeled using all samples and the supervised admixture analysis (Supplementary file 1). The inconsistency of this result across the three analyses leads us to cautiously interpretation of this exception, however.

The assumption that mutations are de novo, when they are not known to occur otherwise in samples investigated from San Salvador Island, should be placed by a more realistic framework. There are certainly many rare alleles that are part of the standing genetic variation, but have previously not been observed. The first time they are seen, may suggest that they are novel mutations.

de novo mutations on San Salvador Island were characterized by their absence in a large genomic sample of pupfishes collected from across the Caribbean and Atlantic, from Massachusetts to Venezuela, including several high-coverage outgroup populations, all described in a previous study (Richards et al., 2021 PNAS). We described these mutations as de novo following the standard use of this term in the population genetic literature and the language used in the initial publication that described these loci as such (Richards et al., 2021, PNAS). We also now include a more detailed description of these novel mutations that addresses the caveat posed by the reviewer and their potential low frequency occurrence on neighboring islands (lines 292-297).

Expressing differences in percent is tricky and potentially open to missunderstandings (e.g. 126 % more accessible). I suggest to use expressions that are more clear (e.g. something like an odd ratio).

We followed this suggestion and now use odds ratios throughout our Results section as requested. Estimates of the odds ratios (as well as their confidence intervals and significance) are also reported in the main figures and tables in the supplementary materials.

Figure 4a. what is the meaning of the different sizes of the coloured dots.

Thank you for catching this. We described their meaning in what was originally Figure 3 (node size is proportional to fitness), but mistakenly omitted the description in the original Figure 4. Node size is now defined in lines 785-786.

Reviewer #2 (Recommendations for the authors):L195-196: I wish the authors could expand a bit on this statement.

Done (addressed throughout lines 214-236).

L 412: "the frequency of rare transgressive hybrid phenotypes was altered" I am not sure what the authors are trying to convey here.

We have clarified this point (lines 520-524).

Figure 2: l180, it should be panel d) and not c).

Fixed.

Citation 66 is a duplicate of 18.

Fixed.

19 papers co-authored by the last author were cited in the current manuscript. While they all seemed relevant, it still is a lot.

We have reduced this to 14 papers. It is difficult to remove more because this study builds heavily on prior work in this system, and we are the only research group presently conducting evolutionary studies on pupfish.

Reviewer #3 (Recommendations for the authors):Line 32: I think whether fitness valleys are crossed is also controversial.

We agree and have reframed these points accordingly (lines 46, 121-129, 432-436).

Line 42: "most isolated" Compared to what? Or is this supposed to say mostly isolated?

Fixed.

Lines 73-74: I think usually "accessible" is used to refer to peaks not trajectories, though I can't say your usage is incorrect.

We use accessible to describe both fitness peaks, as well as genotypic trajectories as per (for example) Weinreich et al., (2006) and Franke et al., (2011). That is, peaks certainly may be described as accessible or not, but only under the condition that accessible genotype trajectories (monotonically increasing in fitness at each mutational step) lead to them.

Line 77: The link between monotonically increasing fitness and plausibility of evolving is much broader than Fisher's geometric model; I think it would be better to just state the simple idea that, in large populations, paths with increasing fitness are taken much more readily than alternatives with neutral or deleterious steps.

We have revised our language as recommended (lines 73-75).

Line 102: Does this mean stable across all these axes of variation?

We now include the axes of variation we are referencing (lake population, year of study, and manipulation of the frequency of rare hybrid phenotypes) – lines 112-114.

Lines 91-110: I'm not clear on how the fitness landscapes in previous work will be distinct from this work, and what parts of Figure 1 support these statements; I suggest a little expansion and reorganization to clarify these issues.

We include at the beginning of this section on the estimation of fitness landscapes a sentence outlining our motivation (lines 238-241). Briefly, the previous studies used slightly different sets of traits that only partially overlapped. Thus, to include all our sequenced samples on a single fitness landscape, we remeasured all individuals for the same suite of traits by a single observer.

Why is Figure 3 cited in the paper before Figure 2?

We have ensured all figures are now cited in the correct order.

I would suggest explicitly defining the fitness measurement in the Results and Discussion, both to make the paper more readable in the current section order and to acknowledge the limitation that reproductive success isn't a part of the composite fitness measurement.

We have made the recommended changes (lines 179-184).